# The respective activation and silencing of striatal direct and indirect pathway neurons support behavior encoding

Christophe Varin [1], Amandine Cornil[1], Delphine Houtteman[1], Patricia Bonnavion[1] & Alban de Kerchove d'Exaerde [1] ✉

The basal ganglia are known to control actions and modulate movements. Neuronal activity in the two efferent pathways of the dorsal striatum is critical for appropriate behavioral control. Previous evidence has led to divergent conclusions on the respective engagement of both pathways during actions. Using calcium imaging to evaluate how neurons in the direct and indirect pathways encode behaviors during self-paced spontaneous explorations in an open field, we observed that the two striatal pathways exhibit distinct tuning properties. Supervised learning algorithms revealed that direct pathway neurons encode behaviors through their activation, whereas indirect pathway neurons exhibit behavior-specific silencing. These properties remain stable for weeks. Our findings highlight a complementary encoding of behaviors with congruent activations in the direct pathway encoding multiple accessible behaviors in a given context, and in the indirect pathway encoding the suppression of competing behaviors. This model reconciles previous conflicting conclusions on motor encoding in the striatum.

The basal ganglia are certainly well known to control both goal-directed behaviors and natural, self-paced behaviors. The proper initiation and execution of these behaviors relies heavily on appropriate functioning within the basal ganglia, as basal ganglia dysfunction is at the core of various disorders, including Parkinson's disease, autism spectrum disorders, and schizophrenia[1]. The striatum, which is the main entry nucleus of the basal ganglia, consists of two types of striatal projection neurons (SPNs) that differ based on their expression of either dopamine D1 or D2 receptors and their respective direct or indirect projections to the output nuclei of the basal ganglia (dSPNs or iSPNs). This functional organization provides differential control of basal ganglia outputs by dSPNs or iSPNs, that leads to net activating or inhibiting effects of thalamocortical circuits, respectively[2,3]. This dichotomy between prokinetic dSPNs and antikinetic iSPNs has been documented using loss-of-function or gain-of-function experiments[4–10], bolstering the traditional go/no-go description of striatal functioning[2,3]. This view has been challenged by correlative descriptions of striatal activity based on recordings of both types of

SPNs, which demonstrated a coactivation of the dSPN and iSPN pathways during locomotion and more generally during actions[11–16]. These results indicate that concerted and cooperative activity between both striatal pathways is needed for proper action initiation and execution.

As a result, two antagonistic models of striatal functioning have been developed that explain how neuronal activity is organized in the striatum. The selection-suppression model[17] postulates that proper action execution relies on the concurrent activation of a small discrete subpopulation of dSPNs that encode the ongoing action and the widespread activation of a large number of iSPNs that inhibit all other actions, with the iSPNs associated with the ongoing action remaining silent. This model predicts that dSPNs are more selective for actions than iSPNs. However, recent investigations highlighted that both pathways encode behaviors with similar properties and dynamics[13–16]. Consequently, a cooperative selection model was proposed[13] in which dSPNs and iSPNs coordinate their activities to select the proper action, with subsets of dSPNs and iSPNs displaying the same targeted activation patterns toward actions. Although this model considers the

[1]Université Libre de Bruxelles (ULB), ULB Neuroscience Institute, Neurophysiology Laboratory, Brussels, Belgium. ✉e-mail: adekerch@ulb.ac.be

coactivation of small ensembles of dSPNs and iSPNs, this model likely fails to take into account the functional opposition between dSPNs and iSPNs[4–10]. In summary, these models predict different patterns of neuronal activation in response to various behaviors, particularly the activity of iSPNs. Therefore, additional investigations are needed to clarify the function of the two SPN pathways as well as their relative organization into functional subpopulations for behavior encoding.

Here, we studied the behavior-encoding properties of dSPNs and iSPNs in the dorsal striatum using one-photon microendoscopy in mice that freely explored an open field and thus expressed a large behavioral repertoire at their own pace. We observed that the behavior-encoding properties of dSPNs and iSPNs differ in a way that challenges the above models. Furthermore, we used support vector machine classifiers to precisely analyze the neural code of dSPNs and iSPNs and their activation patterns during behaviors. We found that, despite their differences in encoding properties, both populations contain the same amount of information to reliably infer behaviors. Moreover, we classified neurons as activated or silent during behaviors to evaluate the predictions of the selection-suppression and cooperative selection models, and we found that neural codes are organized differently in dSPNs and iSPNs. Similar to previous observations in different brain systems, the most important behavior-encoding feature of dSPNs is their specific activation during some behaviors. Remarkably, the most important behavior-encoding feature of iSPNs is their consistent silencing during specific behaviors. Our findings are reinforced by observations that these properties remain stable for weeks.

These results provide the first correlative evidence that dSPNs and iSPNs have distinct encoding properties. Using these observations, we propose an updated model for motor encoding among SPNs in the dorsal striatum. This model relies on the congruent activation of dSPNs, which encode multiple accessible behaviors in a given context to promote these behaviors, and iSPNs, which encode for and inhibit competing behaviors. As a result, the coactivation of specific subsets of behavior-promoting dSPNs and behavior-suppressing iSPNs alongside specific inhibition of subsets of iSPNs allowing behavior expression would result in the selection and execution of only one motor program. This model bridges the gap between various interpretations of experimental observations that promoted antagonistic models on striatal functional organization.

## Results

### Annotation of mice behavior in the open field

To investigate the behavior-encoding properties in both subpopulations in the dorsal striatum, we tracked and reconstructed the behavior of mice freely exploring an open field and simultaneously recorded the neuronal activity of either dSPNs or iSPNs using microendoscopic one-photon imaging of GCaMP6s (Fig. 1a). Mouse self-paced behaviors were identified and labeled using a combination of deep learning tools and clustering methods to generate a predictive model (Fig. 1b and Supplementary Figs. 1 and 2a–g). The workflow for behavior annotation consisted of two parts: an unsupervised detection of behavior clusters, followed by a manual registration of these clusters into one of the 12 behaviors capturing the behavioral repertoire expressed by mice in the open field. In the unsupervised detection of behavior clusters portion, the coordinates of mouse body parts tracked using DeepLabCut[18] were used to compute six features describing the mouse posture and activity (Supplementary Figs. 1 and 2a, b). After multiple iterations of non-linear transformations (t-SNE) to retain postural time series in a low-dimension action space followed by clustering (Gaussian mixture model), the algorithm identified groups of data points consistently clustered together using Hamming distance (on average 88% agreement between partitions; Rand index) forming postural archetypes (Supplementary Fig. 2c). Then these postural archetypes are manually registered into one of the 12 behaviors (Supplementary

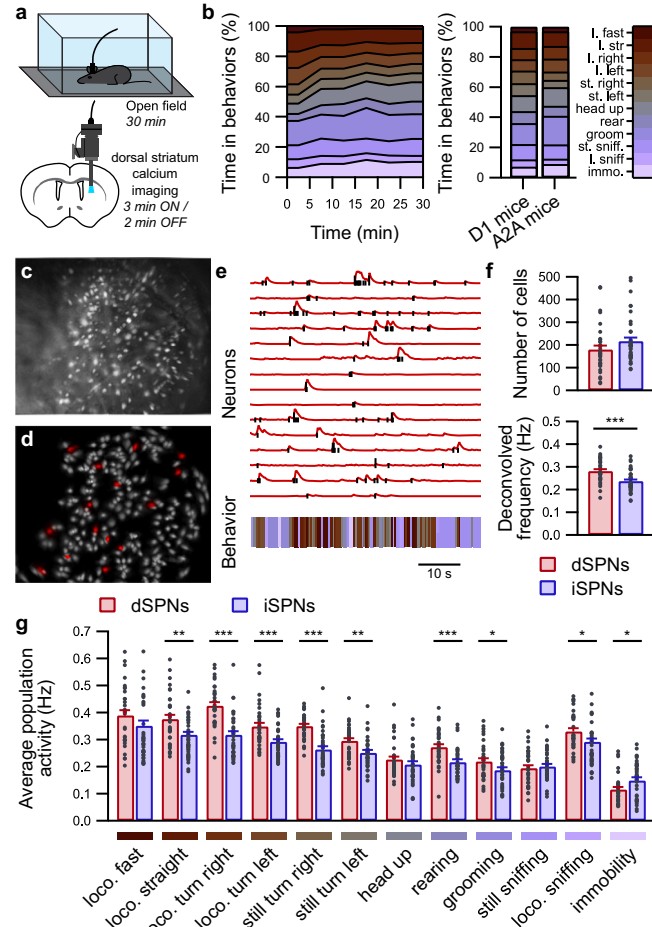

**Fig. 1 | Simultaneous calcium imaging and behavioral identification in freely behaving mice. a** Mice expressing GCaMP6s in either dSPNs or iSPNs and equipped with a microendoscope freely exploring a well-known open field. **b** Temporal evolution of identified behaviors during 30 min of open field exploration for all recording sessions (left panel; 5 min long bins; $n = 73$ sessions in 17 mice) and average distribution of behaviors over 30 min (right panel) in mice expressing GCaMP6s in dSPNs (D1 mice; $n = 33$ sessions with 8 mice) or iSPNs (A2A mice; $n = 40$ sessions in 9 mice). l. locomotion, st. still, immo. immobility. Representative image from one A2A mouse of the maximum fluorescence intensity projection of iSPNs labeled with GCaMP6s (**c**) and the corresponding isolated spatial components identified using CNMF-E (**d**). **e** Representative fluorescence traces (top panel, red lines) and deconvolved calcium activity (top panel, black lines) from selected SPNs (red in **d**) aligned with detected behaviors (bottom panel). **f** Quantification for each recording session of the identified neuron number (top panel) and deconvolved calcium activity (bottom panel) for dSPNs (red; $n = 33$ sessions in 8 mice) and iSPNs (blue; $n = 40$ sessions in 9 mice) (permutation-based two-sided $t$-test, dSPNs vs. iSPNs: number or cells, $p = 0.1012$; deconvolved activity: ***$p = 0$). **g** Average population activity is significantly higher in dSPNs (red; $n = 33$ sessions in 8 mice) than in iSPNs (blue; $n = 40$ sessions in 9 mice) during many behaviors and significantly lower during immobility (linear mixed effect model followed by post hoc permutation-based two-sided $t$-test, dSPNs vs. iSPNs: *$p < 0.05$, **$p < 0.01$, ***$p < 0.001$). loco. locomotion. Data are presented as mean values ± SEM. Detailed statistics are displayed in Supplementary Table 1. Source data are provided as a Source Data file.

Fig. 2d). This second step of manual registration is critical to produce consistent behavior clusters across different mice and across different sessions. The distributions of identified behaviors in the feature space were eventually used to estimate for each time point its likelihood to belong to each behavior cluster (Supplementary Fig. 2e, f). The behavior was determined according to the maximum likelihood (Supplementary Fig. 2f, g).

To test the performance of our behavior identification pipeline, we collected annotation data from four experienced annotators on a common set of 2000 behavior episodes identified by the algorithm. The annotators were exposed twice to the same episodes to evaluate their annotation consistency. Despite behavior definitions reflecting a between-annotator consensus, such a consensus did not prevent variability in annotation style across individuals and within individuals (Supplementary Fig. 2h, i) as evidenced by the intra-annotator comparisons, and comparisons between annotators and annotator #1 or between annotators and the majority classes. Using two complementary metrics evaluating model performance, namely the accuracy and the F1 score, we found that our pipeline achieved performances within the range set by the intra- and inter-annotator agreement (Supplementary Fig. 2i). We observed that our algorithm displayed lower performance to detect head up mostly because it under-detected rearing episodes and mislabeled them as head up events. Similarly, lower performances to detect still sniffing were caused by confusions between still sniffing, grooming, and immobility. Importantly, annotators displayed a similar level of confusion between the same behaviors, which is explained as mice were monitored from above. In general, the behaviors for which our pipeline had the lowest performance coincided with the behaviors for which human annotators had less agreement and less consistency (Supplementary Fig. 2j). Overall, when considering all the behaviors, the average accuracy, precision and recall, and F1 score of our pipeline was comparable to that of human annotators performance (Supplementary Fig. 2k).

Eventually, using our behavior annotation pipeline, we found that behaviors distribution was not homogeneous throughout the course of the open field test, with more locomotor behaviors in the beginning of the session. Additionally, the behavior distributions of the experimental groups were similar (Fig. 1b and Supplementary Fig. 3a–c)

## Behavior encoding properties differ between dSPNs and iSPNs

The calcium activity was extracted from simultaneously recorded microendoscopic images using the CaImAn pipeline[19,20], and the reconstructed temporal traces of calcium activity were deconvolved using MLspike[21] (Fig. 1c–f and Supplementary Figs. 1 and 4a–h). On average, $179 \pm 18$ dSPNs and $216 \pm 16$ iSPNs were identified in each recording session (Fig. 1f). First, we observed that the average population activity of the dSPNs was higher than that of the iSPNs (Fig. 1f). Then, the population activity was decomposed according to the identified behaviors, which revealed that the average population activity was consistently higher for dSPNs than for iSPNs during many behaviors, including straight locomotion, locomotion with right and left turns, remaining still with right and left turns, rearing, grooming, and locomotion sniffing, with the notable exception of immobility, during which the average population activity was significantly lower for dSPNs than for iSPNs (Fig. 1g and Supplementary Fig. 4i, j). This result indicates a substantial difference in the behavioral tuning properties of dSPN and iSPN ensembles.

To better characterize this potential difference among the SPN subpopulations, we first evaluated whether SPN activation was consistent during 30 min of open field exploration. For each behavior, we computed the neuronal activation, which was calculated as the average frequency, during the first and the second halves of each recording, and we evaluated the similarity between these two neuronal activation maps (Fig. 2a). We observed that neuronal activation similarity was higher for dSPNs than for iSPNs for all identified behaviors except grooming, still sniffing, and immobility (Fig. 2b and Supplementary Fig. 5a, b). Similarly, the inverse coefficient of variation is higher for dSPNs than for iSPNs for all behaviors except grooming and immobility (Fig. 2c). These results suggest that, for each behavior, the same dSPNs are more consistently activated, whereas iSPN activation is

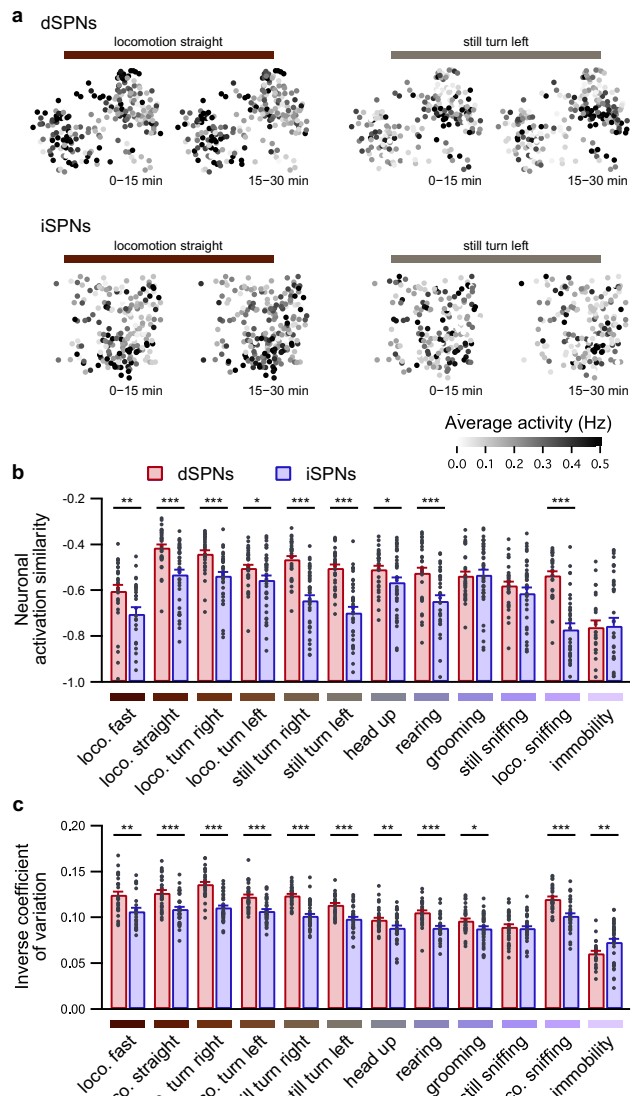

**Fig. 2 | Dynamical properties of behavior encoding differ between dSPNs and iSPNs. a** Representative neuronal activation maps of dSPNs (top panels) and iSPNs (bottom panels) from one session, illustrating the averaged activation of neurons in the first half (0–15 min) and second half (15–30 min) of the recording session for two behaviors (locomotion straight, left; still turn left, right). Note that the neuronal activation appears highly similar during the first and second halves of the recording in dSPNs during the two behaviors and less similar in iSPNs during the same behavior. **b** Neuronal activation similarity between the first and second halves of the open field exploration was higher in dSPNs (red; $n = 33$ sessions in 8 mice) than in iSPNs (blue; $n = 40$ sessions in 9 mice) for all behaviors except grooming, still sniffing, and immobility (linear mixed effect model followed by post hoc permutation-based two-sided $t$-test, dSPNs vs. iSPNs: $^*p < 0.05$, $^{**}p < 0.01$, $^{***}p < 0.001$). **c** Comparison of the inverse coefficient of variation evaluated for each behavior during the whole recording period between dSPNs (red; $n = 33$ sessions in 8 mice) and iSPNs (blue; $n = 40$ sessions in 9 mice) (linear mixed effect model followed by post hoc permutation-based two-sided $t$-test, dSPNs vs. iSPNs: $^*p < 0.05$, $^{**}p < 0.01$, $^{***}p < 0.001$). loco. locomotion. Data are presented as mean values $\pm$ SEM. Detailed statistics are displayed in Supplementary Table 1. Source data are provided as a Source Data file.

inconsistent. Conversely, this observation between the first and second halves of the recordings could result from the fact that both subpopulations are differentially affected by internal drives that accumulate or dissipate during open field exploration, such as stress[22], novelty[23,24], or tiredness[25]. To account for this temporal factor, we

computed the same similarity metric by using all possible partitions of time into two 15-min sets (e.g., 30 min segmented into 5 min long slices). Regardless of the time partition, we observed the same difference in neuronal activation similarity between dSPNs and iSPNs (Supplementary Fig. 5c). Furthermore, the neuronal activation similarity was computed by comparing odd and even frames, by comparing one episode out of two to complementary episodes, and by using the dot product between neuronal activation vectors, yielding the same observations as described above (Supplementary Fig. 6a–c). Moreover, as an additional control to verify the reliability of the similarity measure, we disturbed SPN signaling by artificially increasing dopamine release through acute injection of amphetamine (Supplementary Fig. 7a–d). The neuronal activation similarity of both dSPNs and iSPNs was strongly alleviated after amphetamine administration (Supplementary Fig. 8b). All the above results demonstrate that the difference in neuronal activation similarity is a substantiated time-invariant property of behavior encoding in dSPNs and iSPNs; for each behavior, the same dSPNs are more consistently activated, whereas iSPN activation displays either a milder specificity toward behaviors or is more variable.

Moreover, a comparison of pairs of behaviors revealed that, in the dorsal striatum, similar behaviors are encoded by highly similar neuronal ensembles, whereas dissimilar behaviors are encoded by mildly overlapping neuron groups. Furthermore, this encoding property is equivalent for both SPN subtypes[14]. For each pair of behaviors, we compared the similarity between the average neuronal activity (neuronal activation similarity) to the similarity between behaviors (behavioral similarity). The latter quantifies the similarity in movement trajectories and body shape between each pair of behaviors. We observed a strong positive correlation between pairwise neuronal activation similarity and pairwise behavioral similarity for both dSPNs and iSPNs (Fig. 3a–e). However, when the correlation coefficients of pairwise neuronal and behavioral similarities were compared, we observed a significantly higher correlation for dSPNs than for iSPNs (Fig. 3f). This result indicates that the behavioral space representation differs between the two populations. This result contradicts that of a previous report[14]. To understand this result, we first verified that this difference was not due to the use of a different neuronal similarity measure (Supplementary Fig. 9a). Another key difference is the duration over which animals explored the open field: the animals explored the field for 10–15 min in Klaus et al.[14], whereas our experiments lasted 30 min. Thus, we calculated the correlation coefficients between the neuronal and behavioral similarities for different recording lengths. No difference was detected between dSPNs and iSPNs for durations of up to 10 min (Supplementary Fig. 9b–d). Therefore, extended recordings may be required to collect more samples of neuronal activity during a larger variety of internal or external contexts in order to properly uncover differences in neuronal activation variability for episodes of different behaviors. Overall, these results demonstrate that the coupling between behavior similarity and neuronal similarity is tighter among dSPNs than among iSPNs.

The above observations demonstrate fundamental differences between dSPNs and iSPNs in terms of their dynamical behavior-encoding properties. Moreover, this new set of evidence is inconsistent with existing models of striatal organization. In particular, these models do not account for the existence of different neuronal activation patterns during different episodes of the same behavior. Thus, our results call for a deeper analysis of the neural codes and properties of dSPNs and iSPNs that may convey different behavior-relevant information.

## Behaviors can be reliably decoded from either dSPNs or iSPNs

The first step in our analysis of the neural code in the dorsal striatum was to assess whether animal behaviors can be reconstructed based on instantaneous neuronal activity recorded from either dSPNs or iSPNs.

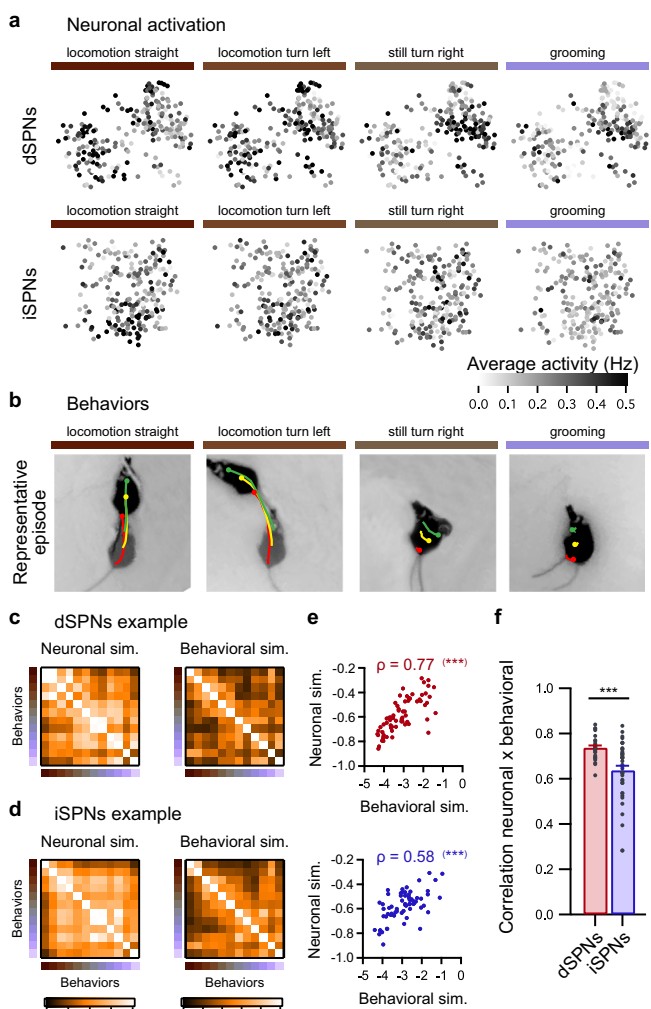

**Fig. 3 | Behavioral space representation differs between dSPNs and iSPNs.** Representative neuronal activation maps of dSPNs (top panels) and iSPNs (bottom panels) during one session for four behaviors (**a**) and representative example of mouse trajectories during one representative episode of the same behaviors (**b**). Examples of matrices of pairwise neuronal activation similarity between behaviors (left panels) and matrices of pairwise behavioral similarity between behaviors (right panels) in one session from one representative dSPNs recording (**c**) and one representative iSPNs recording (**d**). **e** For pairs of behaviors, the neuronal activation similarity and behavioral similarity are significantly correlated in both dSPNs and iSPNs, as illustrated by the examples displayed in (**c**, **d**) (Spearman correlation two-sided: dSPNs, ***$p$ = 9.9e−27; iSPNs, ***$p$ = 3.0e−13). **f** Average correlation coefficient (Spearman correlation) between pairwise behavioral and neuronal similarities is higher for dSPNs (red; $n$ = 33 sessions in 8 mice) than iSPNs (blue; $n$ = 40 sessions in 9 mice) (permutation-based two-sided $t$-test, dSPNs vs. iSPNs: ***$p$ = 0.0005). Data are presented as mean values ± SEM. Source data are provided as a Source Data file.

We thus trained several support vector machine linear classifiers for each pair of behaviors and used the majority rule to combine the outputs of individual binary classifiers (66 individual binary classifiers for each recording) to predict animal behavior (one vs. one multiclass support vector machine) (Fig. 4a). The behavior reconstruction error was calculated for each prediction using the behavioral distance between the predicted behavior and the actual behavior. The instantaneous prediction of behaviors (combined output of binary classifiers) using either dSPN or iSPN activity performs significantly better than chance, which was estimated by decoding behaviors using classifiers trained on time-lagged data. Moreover, dSPNs and iSPNs have similar decoding accuracies and average reconstruction errors (Fig. 4b, c and Supplementary Fig. 16a, b). The prediction accuracy

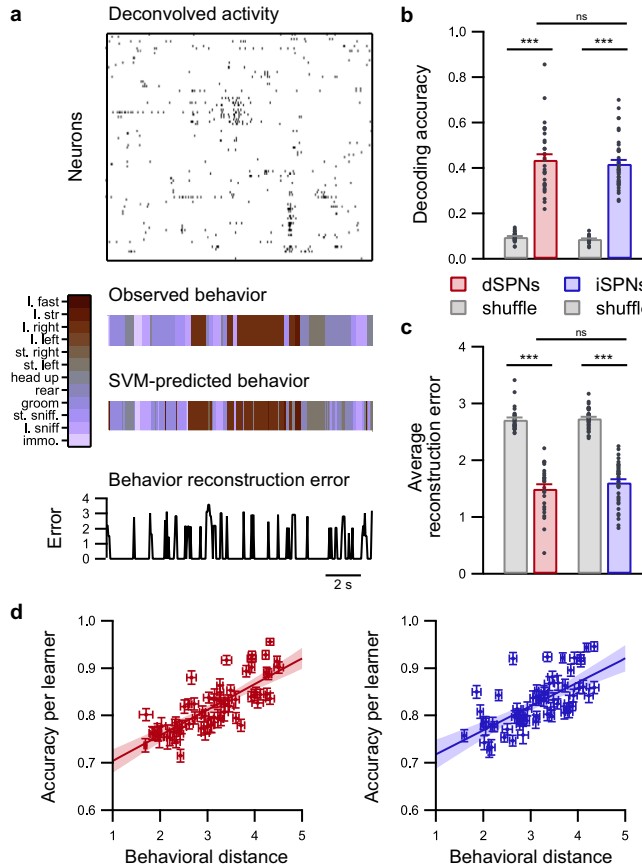

**Fig. 4 | Predicting behaviors is equally efficient when using either dSPN or iSPN ensembles. a** Example of behavior decoding using support vector machines (SVM). Using deconvolved calcium signal (top panel), support vectors are trained to predict ongoing behaviors (middle panel). The prediction error (bottom panel) is calculated by using the behavioral distance between the observed and predicted behaviors. l. locomotion, st. still, immo. immobility. **b** Behavior decoding accuracy was similar when dSPNs (red; $n = 31$ sessions in 8 mice) or iSPNs (blue; $n = 38$ sessions in 9 mice) were used (linear mixed effect model followed by post hoc permutation-based two-sided $t$-test, dSPNs vs. iSPNs: ns $p = 0.292$). Gray, chance level when decoding from time-lagged data (post hoc permutation-based two-sided $t$-test, dSPNs vs. shuffle: ***$p = 0$; iSPNs vs. shuffle: ***$p = 0$). **c** Decoding error evaluated as the average behavioral error for the entire time series using dSPNs (red; $n = 31$ sessions in 8 mice) or iSPNs (blue; $n = 38$ sessions in 9 mice) (linear mixed effect model followed by post hoc permutation-based $t$-test, dSPNs vs. iSPNs: ns $p = 0.137$). Gray, chance level when decoding from time-lagged data (post hoc permutation-based two-sided $t$-test, dSPNs vs. shuffle: ***$p = 0$; iSPNs vs. shuffle: ***$p = 0$). **d** Relationship between the behavioral distance and decoding accuracy for each individual SVM binary classifier using dSPNs (left panel) or iSPNs (right panel). The data for each classifier are presented as the mean (dots) ± SEM across sessions for both the behavioral distance and accuracy. Colored straight lines represent the average regression line, and shaded areas illustrate the 95% confidence interval. Data are presented as mean values ± SEM. Detailed statistics are displayed in Supplementary Table 1. Source data are provided as a Source Data file.

associated with individual binary classifiers (separation between pairs of behaviors) is similar for dSPNs (81.8 ± 1.1%) and iSPNs (82.5 ± 0.8%) ($p = 0.295$). Strikingly, this result appears to contradict previous observations, which indicate differences in behavior encoding between dSPNs and iSPNs. A lower prediction accuracy would be expected when using iSPNs (because of their lower activation similarity) and a larger prediction error would be expected when using iSPNs (because of the lower coupling between pairwise neuronal and behavioral similarities). To confirm the relevance and robustness of our classification strategy, we evaluated whether the decoding

accuracy is affected when SPN activity is strongly disturbed after amphetamine administration. We observed that the decoding performance was considerably reduced following amphetamine administration (Supplementary Fig. 8a, c, d). A closer inspection of the performance of individual classifiers revealed that the decoding accuracy was higher when distinguishing dissimilar behaviors, and lower when distinguishing similar behaviors (Fig. 4d). This property is the same for dSPNs and iSPNs. For example, for dSPNs, the separation between considerably different behaviors, such as locomotion turn right and immobility (behavior distance: 3.93 ± 0.06), is more accurate (accuracy: 90.2 ± 1.1%) than the separation between more similar behaviors, such as locomotion turn right and locomotion turn left (behavior distance: 1.79 ± 0.06; accuracy: 76.1 ± 1.3%). These results indicate that despite previous evidence on differences in dynamical behavior encoding, both striatal populations contain and encode the same amount of information in response to behaviors.

## Neural code in dSPNs is biased toward activation
To better characterize the neural code in the dorsal striatum, we attempted to identify which features in the response properties of individual cells contribute most to behavior encoding.

We first identified which neurons were significantly activated during specific behaviors and defined these neurons as behavior-active when the mutual information between the behavior time series and the recorded neuronal activity (behavior information) was statistically significant[26,27] (Fig. 5a, b and Supplementary Fig. 10). Based on this criterion, the proportion of behavior-active cells is higher among dSPNs than among iSPNs (Fig. 5b and Supplementary Fig. 11a). Interestingly, the proportion of behavior-active dSPNs is significantly larger for contralateral turns (right turns) than ipsilateral turns during locomotion or without locomotion (Supplementary Fig. 11a) (dSPNs; loco. turn right vs. loco. turn left, permutation-based $t$-test, $p = 0$; still turn right vs. still turn left, permutation-based $t$-test, $p = 0$). Similarly, the proportion of behavior-active iSPNs is larger for contralateral turns than ipsilateral turns during locomotion (iSPNs; loco. turn right vs. loco. turn left, permutation-based $t$-test, $p = 0.0006$), whereas no difference was observed between turn directions without locomotion (iSPNs; still turn right vs. still turn left, permutation-based $t$-test, $p = 0.079$). According to this classification of whether individual neurons are behavior-active, we evaluated whether behaviors could be decoded based on the activity of behavior-active or non-behavior-active neurons. For dSPN recordings, the prediction was significantly more efficient when using behavior-active cells than when using non-behavior-active cells, whereas, for iSPN recordings, the prediction performance was similar when either behavior-active cells or non-behavior-active cells were used (Fig. 5c, d and Supplementary Figs. 11b, c and 16c−f). Moreover, the decoding performance was similar when non-behavior-active dSPNs or iSPNs were used, whereas the decoding performance was significantly higher when behavior-active dSPNs were used than when behavior-active iSPNs were used (Fig. 5c, d). This observation was maintained when the level of significance of behavior information for classifying neurons as behavior-active was changed (Supplementary Fig. 11d). The same results were observed when significantly activated neurons were detected using a shuffle procedure for each behavior independently (Supplementary Fig. 12). Taken together, these findings demonstrate that the neural code for behaviors is biased toward behavior-active dSPNs, whereas this feature is more evenly distributed among iSPNs.

## Neural code in iSPNs is biased toward silencing
According to the selection-suppression model[17], indirect pathway SPNs are activated to suppress all motor programs except the one being executed. As a result, iSPNs responsible for suppressing a particular behavior are never active during this behavior. Thus, following this postulate, we investigated the relative contribution of neurons that remain silent during behaviors to the neural code. We thus

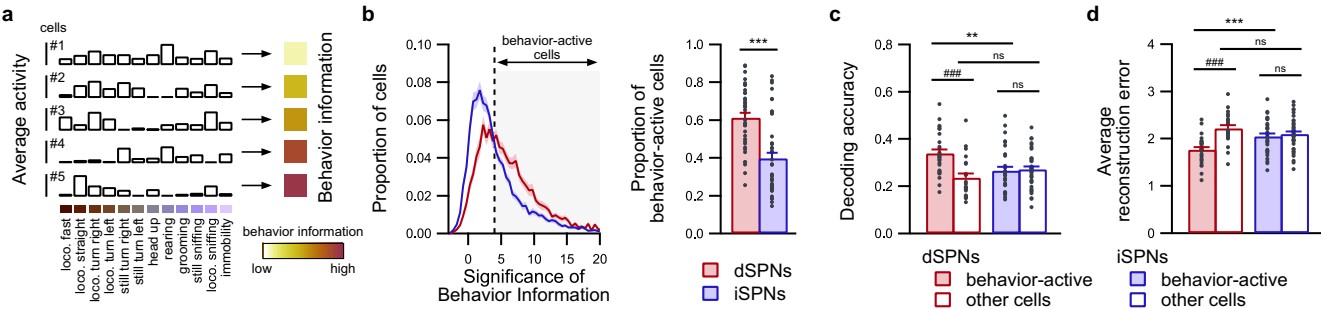

**Fig. 5 | Relevant information for decoding behaviors lies primarily in excited dSPNs. a** Information content of neuronal activity in response to behaviors was derived based on the behavior information, which was calculated as the mutual information between the behavior time series and neuronal activity. As illustrated for 5 representative neurons, a higher level of behavior information reflects a stronger degree of tuning of neuronal activation toward behaviors. loco. locomotion. **b** Cells are identified as behavior-active when the level of significance of behavior information exceeds 4 sigma of the shuffled distribution (gray area; left panel; threshold indicated by the black dashed vertical line). A larger proportion of behavior-active cells was identified among dSPNs ($n = 29$ sessions in 8 mice) than iSPNs ($n = 37$ sessions in 9 mice) (right panel; permutation-based two-sided $t$-test,

dSPNs vs. iSPNs: ***$p = 0$). Decoding accuracy (**c**) and average decoding error (**d**) when predicting behaviors using behavior-active neurons (plain bars) and non-behavior-active cells (unfilled bars). The decoding performance is better for behavior-active cells than for non-behavior-active cells in dSPNs recordings (red bars; $n = 29$ sessions in 8 mice), whereas it is similar in iSPNs recordings (blue bars; $n = 37$ sessions in 9 mice) ensembles. Moreover, the decoding performance is better for behavior-active dSPNs than for behavior-active iSPNs (linear mixed effect model followed by post hoc permutation-based two-sided $t$-test, dSPNs vs. iSPNs: ns $p > 0.05$, **$p < 0.01$; behavior-active vs. other cells: ns $p > 0.05$, ###$p < 0.001$). Data are presented as mean values ± SEM. Detailed statistics are displayed in Supplementary Table 1. Source data are provided as a Source Data file.

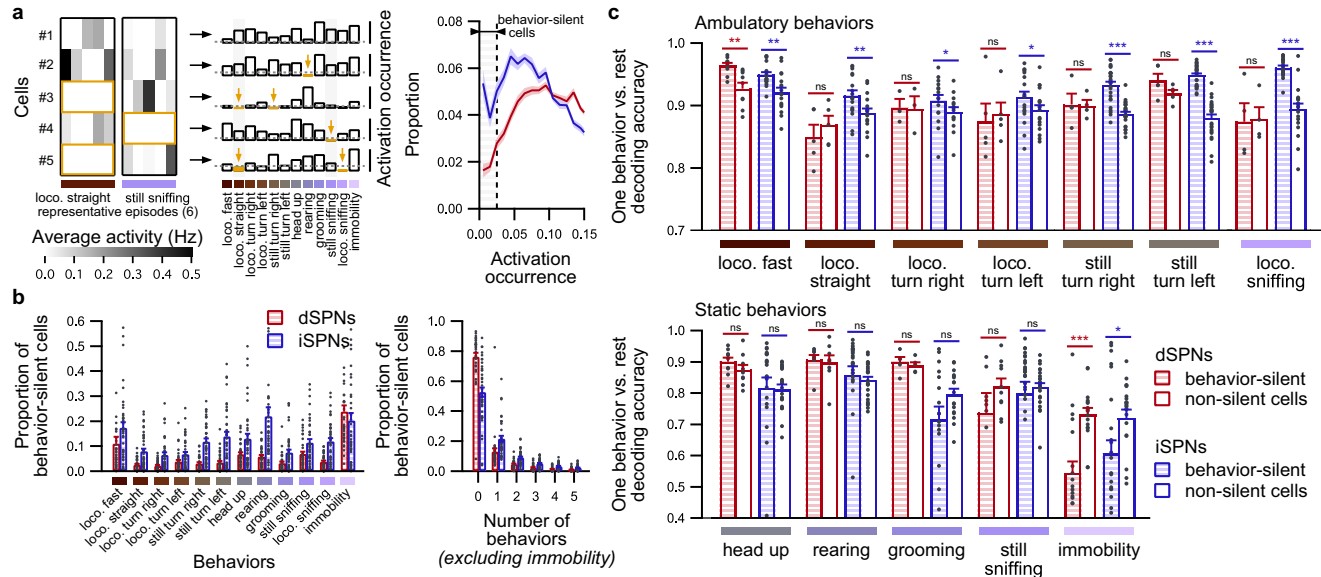

**Fig. 6 | Relevant information for decoding behaviors lies primarily in silent iSPNs. a** Behavior-silent cells are identified according to how often these cells are active during episodes of a given behavior, as illustrated by 5 representative neurons (left panel; 6 representative episodes displayed) that displayed various levels of activation occurrence for each behavior (middle panel). Some neurons are rarely or never active during all episodes of a given behavior (middle panel, yellow bars and arrows). The activation occurrence threshold was set to 0.025 according to the distribution of activation occurrence values for all behaviors in dSPNs and iSPNs (right panel; threshold indicated by the black dashed vertical line). **b** Average fraction of dSPNs (red; $n = 33$ sessions in 8 mice) and iSPNs (blue; $n = 40$ sessions in 9 mice) identified as behavior-silent for each behavior (left panel) and average fraction of dSPNs and iSPNs categorized as behavior-silent during no behavior or

one to five behaviors (right panel). **c** One behavior vs. rest decoding accuracy (simple matching coefficient) for prediction in separating each behavior from other behaviors using neurons that were classified as behavior-silent during this behavior (hatched bars) or non-silent (unfilled bars) in dSPNs (red; $n = 29$ sessions in 8 mice) or iSPNs recordings (blue; $n = 37$ sessions in 9 mice) (linear mixed effect model followed by post-hoc permutation-based two-sided $t$-test, behavior-silent vs. non-silent neurons: ns $p > 0.05$, *$p < 0.05$, **$p < 0.01$, ***$p < 0.001$). Note that the decoding performance is higher for behavior-silent iSPNs than for non-silent cells during ambulatory behaviors (top panel), whereas this result is not observed during static behaviors (bottom panels). loco. locomotion. Data are presented as mean values ± SEM. Detailed statistics are displayed in Supplementary Table 1. Source data are provided as a Source Data file.

identified SPNs that are consistently silent during episodes of each behavior by computing, for each neuron, how often this neuron is active during all episodes of this behavior (Fig. 6a). For each behavior, if this activation occurrence is sufficiently low (threshold of 2.5% of episodes; Fig. 6a), the neuron is labeled as behavior-silent for this behavior (Supplementary Fig. 13). Strikingly, the proportion of behavior-silent neurons was higher in iSPNs than in dSPNs for each

behavior (Fig. 6b), with the notable exception of immobility, for which the proportion was similar in dSPNs and iSPNs. Then, we separated SPNs classified as behavior-silent or non-behavior-silent and evaluated the information content of these groups of cells to accurately separate the associated behavior from other behaviors (one behavior vs. rest decoding accuracy) (Fig. 6c). We chose this measure because, according to the selection-suppression model[17], consistent silencing in

some iSPNs should indicate that the animal is currently engaged in a given behavior, whereas their activation, leading to the suppression of this given behavior, should indicate that the animal experiences any other behavior without enabling to directly identify this other behavior. For dSPNs, the separation accuracy of all behaviors was similar when behavior-silent cells or non-behavior-silent cells were used, except fast locomotion and immobility. On the other hand, we observed that the separation accuracy was consistently higher for behavior-silent iSPNs than non-behavior-silent iSPNs during ambulatory behaviors (i.e., behaviors involving locomotion or right and left turns without locomotion), indicating that silencing is an important feature of the neural code in iSPNs during these behaviors. Conversely, this difference in separation accuracy between behavior-silent and non-behavior-silent iSPNs was not observed for static behaviors (i.e., head up, rearing, grooming, still sniffing, and immobility) (Fig. 6c and Supplementary Fig. 16g–i). This distinction between ambulatory and static behaviors could be because the detected static behaviors may contain additional substates that were not separated in our classification. Furthermore, this distinction may represent different modes of encoding in iSPNs during ambulatory and static behaviors, such as spatial mapping in the dorsal hippocampus, which is characterized by place cells that display their specific place fields mostly during locomotion[28,29]. Additionally, similar results were observed for iSPNs when the classification relied on the comparison of activation occurrence to the distribution of activation occurrences obtained from random permutations (Supplementary Fig. 14a–d). We also observed the same results for both dSPNs and iSPNs when we used an alternate cell classification, which was based on a threshold on the average activity during behaviors (Supplementary Fig. 15a–c). In conclusion, these observations support a key distinctive feature of iSPNs for behavior encoding, namely, that iSPNs are consistently silent during behaviors, a property that is not observed for dSPNs.

## Long-term stability of neural code over weeks

To reinforce our previous findings and assess their reliability, we investigated their long-term stability. Thus, for each animal, we performed a longitudinal registration of neurons[30] across pairs of recording sessions over 1 month (Supplementary Figs. 1 and 17a). The registration performance was similar for dSPN and iSPN recordings (Supplementary Fig. 17b). On average, $104 \pm 5$ cells were recovered between any pair of sessions (Supplementary Fig. 17c), corresponding to an overlap of $33.9 \pm 0.8\%$ in the subset of cells identified in both sessions, ranging from $36.2 \pm 1.5\%$ after ~5–7 days to $32.5 \pm 1.4\%$ after ~30 days. As described above, we first quantified the neuronal activation similarity for each behavior between pairs of sessions. For all behaviors, the neuronal activation similarity was higher for dSPNs and iSPNs than for their respective shuffles, as evaluated by either random permutations of registered cell pairs or by replacing one cell or each pair with its closest neighbor (Fig. 7a and Supplementary Fig. 17d, e). These controls provide an additional post-hoc validation of the longitudinal registration procedure. Then, comparisons between dSPNs and iSPNs revealed that, with the exception of immobility, the neuronal activation similarity was consistently higher over time for dSPNs than for iSPNs (Fig. 7a and Supplementary Fig. 17d, e). This result appears to indicate that neuronal activation patterns are preserved better for dSPNs than for iSPNs, which may reflect the previously established bias toward activation in the neuronal code of behaviors in dSPNs.

Following this idea, we evaluated the stability of the behavior-coding properties across sessions by comparing the classification of cells into the behavior-excited or behavior-silent categories using the Jaccard index (intersection over union) for binary attributes. We observed that within registered cells between pairs of sessions, the proportion of neurons that remained classified as behavior-active for the same behaviors was higher among dSPNs than among

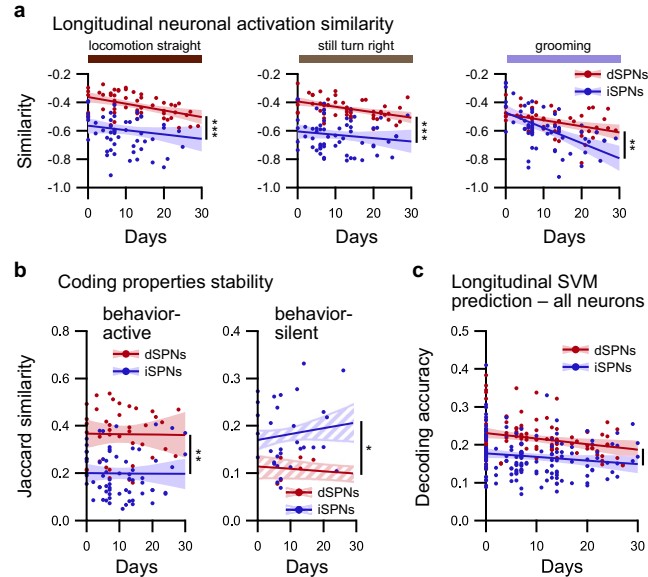

**Fig. 7 | Long-term stability of the respective coding properties of dSPNs and iSPNs. a** Differences in neuronal activation similarity during identified behaviors between dSPNs (red; $n = 52$ pairs of sessions in 8 mice) and iSPNs (blue; $n = 62$ pairs of sessions in 9 mice) are preserved for weeks, as illustrated for locomotion straight (left), still turn right (middle), and grooming (right) (linear mixed effect model followed by post hoc analysis of covariance, dSPNs vs. iSPNs: ***$p < 0.001$). **b** Quantification of the long-term stability of coding properties of neurons using the Jaccard similarity coefficient between binary classification of longitudinally registered neurons that were labeled as behavior-active (left panel) or behavior-silent during ambulatory behaviors (right panel) in dSPNs (red; behavior-active: $n = 58$ pairs of sessions meeting criterion; behavior-silent: $n = 16$ pairs of sessions meeting criterion) and iSPNs recordings (blue; behavior-active: $n = 70$ pairs of sessions meeting criterion; behavior-silent: $n = 34$ pairs of sessions meeting criterion) (linear mixed effect model followed by post hoc analysis of covariance, dSPNs vs. iSPNs: *$p < 0.05$, **$p < 0.01$). Note that the similarity is higher for behavior-active dSPNs and behavior-silent iSPNs. **c** Accuracy in predicting the behavior according to the activity of longitudinally registered neurons using SVM classifiers trained on different recording sessions for dSPNs (red; $n = 121$ pairs of sessions) and iSPNs (blue; $n = 171$ pairs of sessions) (linear mixed effect model followed by post hoc analysis of covariance, dSPNs vs. iSPNs: **$p < 0.01$). Data are presented as mean values $\pm$ SEM. Detailed statistics are displayed in Supplementary Table 1. Source data are provided as a Source Data file.

iSPNs (Fig. 7b). Conversely, during ambulatory behaviors, behavior-silent iSPNs overlapped between sessions more often than behavior-silent dSPNs (Fig. 7b and Supplementary Fig. 18c). This result indicates that the classification of dSPNs as behavior-active and the classification of iSPNs as behavior-silent are more consistent over time than their respective counterparts in the other striatal population.

To deepen this observation on the categorization of single neurons and extend it to neural ensembles, we quantified whether support vector machines trained on neuronal activity and behavior time series on a given day can efficiently predict the behavior using the neuronal activity of the same neurons on a different day. For both dSPNs and iSPNs, the longitudinal prediction of behaviors using support vector machines is consistently more accurate than the predictions obtained with classifiers trained on time-lagged data (Fig. 7c and Supplementary Fig. 18a, b). Interestingly, the longitudinal prediction of behaviors performs better with dSPNs than with iSPNs, which might reflect the higher long-term stability of neuronal activation similarity for dSPNs.

Finally, we analyzed whether the activity of dSPNs and iSPNs categorized as behavior-active or behavior-silent on a given day can be used to efficiently predict behaviors on a different day. We observed that for dSPNs, behavior predictions were significantly more accurate

when only behavior-active cells were used than when non-behavior-active cells were used (Fig. 8a). In contrast, the accuracy is similar when behaviors are predicted with either behavior-active iSPNs or non-behavior-active iSPNs (Fig. 8a). This finding indicates that the bias of the neural code toward behavior-active dSPNs and the information content of these cells in response to behaviors remain conserved for weeks. For behavior-silent neurons (or behavior-inactive neurons, using the alternate definition), we noted that the prediction of ambulatory behaviors is significantly more accurate with behavior-silent iSPNs than with non-behavior-silent cells (Fig. 8b and Supplementary Fig. 18d), whereas no difference was observed for static behaviors. On

the other hand, there is no difference in the prediction accuracy when either behavior-silent or non-behavior-silent dSPNs are used (Fig. 8b and Supplementary Fig. 18d). This finding highlights the long-term preservation of the bias toward behavior-silent iSPNs in the striatal neural code. Overall, these results demonstrate the long-term stability of the neuronal encoding properties established in individual recording sessions, reinforcing the reliability of our findings.

Thus, our findings demonstrate that neuronal representations of spontaneous self-paced behaviors in striatal ensembles are biased toward activation in dSPNs and silencing in iSPNs and that these representations are preserved for weeks. These results provide new original insights into both the neuronal organization of striatal ensembles for behavior encoding and efficient motor program selection. In addition, our observations allow us to propose an updated model for behavior encoding in the two striatal pathways that solves discrepancies raised by previously formulated hypotheses.

## Discussion

Neurons in the dorsal striatum exhibit diverse and heterogeneous responses to different external variables[11,13,14,31]. These responses cannot be easily interpreted and highlight the challenge of identifying the distinct encoding properties of the two parallel efferent pathways in the dorsal striatum. In this study, we evaluated the neuronal encoding properties of SPN ensembles during self-paced natural behaviors in an open field. In this experimental paradigm, animals can express unconstrained naturalistic behaviors forming a large behavioral repertoire. In our experiments, we observed previously unreported differences between direct and indirect pathway SPNs. These differences most likely reflect the higher variability of iSPNs in neuron ensembles that are activated during different episodes of the same behavior, which could be observed only during long sessions of open field exploration. However, despite these heterogeneous response properties, the behaviors could be reliably decoded from the activity of either dSPNs or iSPNs, demonstrating that both pathways contain the same level of information with respect to the ongoing behaviors. Therefore, it questions the respective organization of the neural code in both pathways.

Many formulations have been proposed to explain the organization of neuronal activity in the two striatal pathways. Among them, a "complete selection-suppression" model proposes prokinetic and antikinetic functions for the direct and indirect pathways, respectively[4–6,17] (Supplementary Fig. 19a). This model proposes that proper motor program execution relies on the congruent activation of a small discrete subpopulation of dSPNs that encode this motor program and a large number of iSPN ensembles associated with all other motor programs that inhibit all other behaviors. Such a model predicts that dSPNs are more selective for actions than iSPNs, which is inconsistent with observations in previous reports[11,14–16] and our results. However, this model is compatible with our observation that the neural code among dSPNs is biased toward activation, whereas the neural code is biased toward inhibition in iSPNs because during a given behavior, dSPNs associated with this behavior are activated, whereas iSPNs associated with the same behavior remain silent (Supplementary Fig. 19d). On the other hand, the "cooperative selection" model[13] (Supplementary Fig. 19b) proposes the cooperative activation of discrete dSPNs and iSPNs ensembles that are similarly tuned toward behaviors to select proper motor programs. Although this model accurately incorporates the coactivations of similar size dSPN and iSPN ensembles during actions, it does not account for the functional dissimilarities of dSPNs and iSPNs[4–6] or our observations of dissimilar encoding dynamics between the two striatal pathways, in particular the bias toward silencing in iSPNs (Supplementary Fig. 19d). As a result, the current models of striatal organization must be reevaluated. This reevaluation needs to take into account our novel observations. Here, we report for the first time divergent behavior-encoding properties

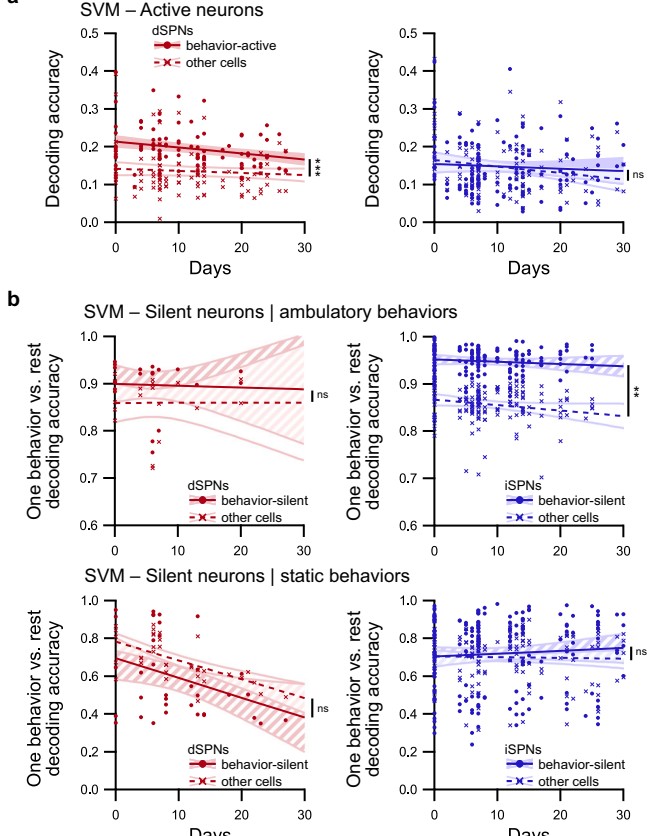

**Fig. 8 | Long-term prediction of behaviors using behavior-active dSPNs and behavior-silent iSPNs. a** Decoding accuracy for longitudinal predictions of behaviors using behavior-active neurons (circles, plain line, colored confidence interval) or non-behavior-active cells (crosses, dashed line, unfilled confidence interval) in dSPNs (top panel; red; $n = 93$ pairs of sessions meeting criterion) or iSPNs recordings (bottom panel; blue; $n = 132$ pairs of sessions meeting criterion). The accuracy remained higher for behavior-active dSPNs than non-behavior-active dSPNs for many days, whereas the accuracy was similar for both groups of iSPNs (linear mixed effect model followed by post hoc analysis of covariance, behavior-active vs. other cells: ns $p > 0.05$, ***$p < 0.001$). **b** Simple matching coefficient for long-term predictions on separating one behavior from other behaviors using neurons classified as behavior-silent (circles, plain line, hatched confidence interval) or non-silent (crosses, dashed line, unfilled confidence interval) during this behavior and pooled for ambulatory behaviors (top panels) and static behaviors (bottom panels) for dSPNs (left panels; red; ambulatory: $n = 18$ pairs of sessions meeting criterion; static: $n = 46$ pairs of sessions meeting criterion) and iSPNs (right panels; blue; ambulatory: $n = 52$ pairs of sessions meeting criterion; static: $n = 69$ pairs of sessions meeting criterion). Note that the only difference between behavior-silent and non-behavior-silent cells is observed for iSPNs during ambulatory behaviors (linear mixed effect model followed by post hoc analysis of covariance, behavior-silent vs. other cells: ns $p > 0.05$, **$p < 0.01$). Data are presented as mean values ± SEM. Detailed statistics are displayed in Supplementary Table 1. Source data are provided as a Source Data file.

between dSPNs and iSPNs, which may reflect their functional dissimilarity. In particular, our observation that the same dSPNs are more consistently activated for each behavior, whereas iSPNs activation is more inconsistent, reveals that behavior specificity is milder in iSPNs than in dSPNs. This consideration is compatible with the general framework of the "complete selection-suppression" model. Additionally, one aspect of our recordings that should be considered is the high variability in neuronal activation in both dSPNs and iSPNs for different episodes of the same behavior. This feature has not been integrated into current models and likely reflects the large effect of external and internal contexts for all occurrences of a given motor program. We thus propose a novel formulation we call "adaptive selection-suppression" (Supplementary Fig. 19c), which attempts to reconcile the abovementioned models, previously reported observations, and our observations (Supplementary Fig. 19d). We hypothesize that dSPNs encode a small subset of accessible behaviors in the overall behavioral repertoire, including the observed behavior, and are highly dependent on the ongoing context and activated to promote these behaviors. Furthermore, specific iSPNs that encode competing behaviors, which are also highly dependent on the context, are activated to suppress these competing behaviors. In the subsequent basal ganglia nuclei, the comparison between selecting dSPN activations and suppressing iSPN activations dictates the ongoing motor program. This model supports our observation that similar behaviors correspond to comparable neuronal activations in dSPNs and iSPNs, as adjacent, highly similar behaviors could be more often competing with each other than considerably different behaviors. However behavior similarity is probably a poor estimator of the degree of competition between behaviors and a further understanding of this phenomenon would require a proper dedicated measure of behavior competition. Moreover, an important feature of the neural code in this model is that specific subgroups of dSPNs are activated in response to the expressed motor program, whereas specific subgroups of iSPNs are consistently inactive. This finding is substantiated by our observation that the most relevant information for predicting behaviors is located in dSPNs that are activated during behaviors and iSPNs that remain silent during behaviors. This model guarantees the coactivation of discrete subsets of dSPNs and iSPNs during each behavior and supports the broad prokinetic and antikinetic functions of the direct and indirect striatal pathways, respectively. Alternatively the continuous context-dependent representation of the action space in SPNs could reflect the propensity or the value associated with the execution of possible movements on a more abstract level gathered in broad action categories instead of highly defined complex behaviors[14,15,32–34]. Further studies investigating the neuronal activity in the dorsal striatum and in downstream nuclei are necessary to clarify whether the specification of detailed actions occurs in the striatum or downstream in the basal ganglia.

As described above, for each observed motor program, our model incorporates the notion of inter-episode variability, which depends on the context dictated by both internal and external factors. Indeed, the dorsal striatum incorporates different representations of various contextual modalities, such as spatial information[31,35], visual and tactile cues[36], timing[37], rewarding and aversive drives[38], task constraints[15], and sleep drive[25]. These studies substantiate the idea of rich, highly context-dependent behavior representations supported by SPNs. These representations likely originate in cortical areas[39,40] and are transferred to the dorsal striatum for subsequent processing and integration[41–43]. Moreover the information about behaviors and contextual modalities is probably highly distributed across individual neurons. Our analyses show that non-behavior-active or non-behavior-silent cells contribute to the encoding of behaviors (e.g., decoding behaviors using non-behavior-active dSPNs performs better than chance). This observation may be explained by the fact that, although some neurons, when evaluated individually, encode behaviors weakly

(do not pass the criterion for being categorized as behavior-active or behavior-silent), these "untuned" cells may contribute to behavior encoding through shared ensemble activity patterns or because of their correlations with behavior-active or behavior-silent SPNs. A contribution of these "untuned" neurons to the neural code was identified, for example, in the hippocampal formation or in the prefrontal cortex[26,44,45]. In addition, our model proposes that dSPNs are activated not only in response to the expressed behavior but also in response to additional, often similar behaviors. The concurrent activation of specific iSPN ensembles enables the proper filtering of only one action, most likely the most appropriate action in a given context. This mode of organization likely supports efficient shifts in the behavioral strategy depending on an animal's internal state or task contingencies[46].

In addition, in our study, we did not observe a bias toward silencing among iSPNs during static behaviors. This could be because these behaviors contain supplemental substates that we could not separate with our analysis pipeline. Alternatively, this could be a consequence of different modes of encoding in iSPNs during ambulatory and static behaviors, similar to spatial mapping in the dorsal hippocampus, which is prominent mainly during locomotion[28,29]. This alternation between encoding modes may rely on functional interactions between the dorsal hippocampus and the dorsal striatum[47,48].

Furthermore, we demonstrated the long-term stability of the neural code in SPNs. We observed that behavior-related information was retained for weeks in behavior-active dSPNs and behavior-silent iSPNs, reinforcing our findings. More precisely, behavior-coding ensembles display a fluctuating membership that ultimately preserves behavior-related information. This finding supports the idea of stable behavioral representations in the dorsal striatum that exhibit some day-to-day fluctuations at the single-cell level. This coding turnover could preserve a certain level of flexibility within the system, which may enable the formation of new traces for encoding similar motor programs that occur in different environments or varied contexts. This mechanism could be critically engaged during procedural or episodic memory formation[49].

In summary, we identified clear functional differences between SPNs in the direct and indirect pathways in the dorsal striatum in response to self-paced spontaneous behaviors. Our data indicate that ongoing behaviors can be decoded based on dSPN and iSPN activity patterns, despite the fact that these patterns resemble those associated with similar behaviors. Behavior-specific firing and silencing are more prominent in dSPNs and iSPNs, respectively. Our observations are consistent with a model in which dSPN activations represent the ongoing behavior alongside competing motor programs, while iSPNs specific for the ongoing behavior become silent and iSPNs specific for competing behaviors become active. These observations are critical for a deeper understanding of striatal functional organization and strengthen the view that direct and indirect pathways cooperatively orchestrate motor programs in a manner that is highly dependent on the ongoing context by selecting a small subset of accessible behaviors while suppressing competing behaviors.

## Methods

### Animal care and use

All procedures were performed according to the Institutional Animal Care Committee guidelines and were approved by the Local Ethical Committee (Comité d'Ethique et de Bien-Être Animal du pôle santé de l'Université Libre de Bruxelles (ULB), Ref. No. 646 N). Mice were maintained on a 12-h dark/light cycle (lights on at 8 pm) with ad libitum access to food and water. The room temperature was set to $22 \pm 2\,°C$ with constant humidity (40–60%). The behavioral tests were performed during the dark photoperiod. Both male and female transgenic mice (≥8 weeks) were used in all behavioral experiments.

### Transgenic mouse generation

The genetic background of all transgenic mice used in this paper is C57Bl/6J. The mice were heterozygous and maintained by breeding with C57Bl/6 mice. All lines were backcrossed with C57Bl6 mice for at least 8 generations. Three transgenic mouse lines were used: $A_{2A}$-Cre[7], $D_1$-Cre (EY262; GENSAT)[50] and Ai162/TIT2L-GC6s-ICL-tTA2 (Ai162D; Allen Institute)[51]. Simple transgenic $A_{2A}$-Cre or $D_1$-Cre mice (A2A(AAV) and D1 mice, respectively) were used for the virally mediated targeting of iSPNs or dSPNs, respectively. Double transgenic $A_{2A}$-Cre x Ai162/TIT2L-GC6s-ICL-tTA2 mice (A2A(Tg) mice) were generated by breeding and used for targeting iSPNs without virus injection. For this breeding, mice were maintained with doxycycline food pellets (A03 1 g/kg doxycycline hyclate pellet; SAFE, France) to prevent GCaMP6s expression during development and early life stages in offspring. Standard food was introduced when offspring were weaned (3 weeks postnatal). The results of $A_{2A}$-Cre mice expressing GCaMP6s through a virus-mediated strategy (A2A(AAV) mice) or through a double transgenic strategy (A2A(Tg) mice) were compared, and no differences were observed (Supplementary Fig. 4d, e, h). As a consequence, these animals were pooled.

### Viral injections and chronic lens implantation

Under isoflurane anesthesia (induction 4%, maintenance 1% in $O_2$; 0.5 l/min), male and female $A_{2A}$-Cre and $D_1$-Cre mice (≥8 weeks old), which targeted iSPNs and dSPNs, respectively, received two injections (500 nl per site at 100 nl/min) under stereotaxic control in the left dorsal striatum (AP: +1.2 mm; ML: −1.75 mm; DV: −3 mm; and AP: +1.2 mm; ML: −1.75 mm; DV: −3.3 mm relative to Bregma according to the Franklin and Paxinos atlas third edition) of a Cre-dependent virus encoding GCaMP6s (AAV-Dj-EF1α-DIO-GCaMP6s; Stanford Vector Core, titer $5.02 \times 10^{12}$ vg/ml; UNC Vector Core, titer $3.9 \times 10^{12}$ vg/ml), which was delivered through a cannula connected to a Hamilton syringe (10 µl) placed in a syringe pump (KDS-310-PLUS, KDScientific). Cannulas were lowered into the brain and left in place for 10 min after infusion. Mice (4–5 weeks after virus injection or ≥8 weeks old for A2A(Tg) mice) were then prepared for in vivo calcium imaging. A gradient index (GRIN) lens (1.0 mm or 0.6 mm diameter, ID-1050-004605 or ID-1050-004608; Inscopix, Palo Alto, CA) was implanted into the left dorsal striatum under stereotaxic control directly above the injection site (AP: +1.2 mm; ML: −1.9 mm; DV: −2.8 mm relative to Bregma). Once the lens was positioned, the lens was secured to the skull using Metabond (C&B, Sun Medical Co. Ltd, Japan) and protected using tape. Two weeks after lens implantation, a microendoscope baseplate (ID-1050-004638; Inscopix) was attached to the skull with Metabond in the optimal imaging plane (550 µm above the lens for the 0.6 mm diameter lens and ~300 µm above the lens for the 1.0 mm diameter lens). The behavioral experiments began at least 1 week after the baseplate was fixed to ensure that the field of view was adequately cleared.

### Open field behavior

The behavior experiments were conducted in an open field arena (40 cm × 40 cm × 40 cm, length × width × height) with white walls in a dark environment (0 lux). Before the first recording session, mice were habituated to the open field and the microendoscope for at least 5 consecutive days by using a dummy microscope (ID-1050-003762; Inscopix) mounted in place of the actual microscope. The animal behavior was recorded for 30 min for 4–5 sessions spaced over 5–7 days using a camera (sampling rate: 40 fps) mounted on the ceiling ~1.5 m above the arena controlled through EthoVision XT14 (Noldus). One photon calcium imaging was sampled at 20 fps using a nVista 3.0 microendoscope (Inscopix) controlled through Inscopix Data Acquisition Software (IDAS, versions 1.2.1 to 1.5.1; Inscopix). A commutator (Inscopix) attached to the ceiling was placed between the camera and the acquisition box to minimize cable entanglement.

To prevent photobleaching, calcium frames were acquired with a 2 min OFF/3 min ON pattern, and time synchronization between the calcium recordings and open field videos was programmed and managed through EthoVision XT14. Proper alignment between behavior videos and calcium imaging videos was ensured thanks to the recording by the Inscopix DAQ box of the trigger TTL signal emitted by EthoVision XT14 and the recording by Ethovision XT14 of the status of the Inscopix microscope generated by the Inscopix DAQ box. The comparison between both signals enabled to correct for any delay or differences in acquisition rate between behavior videos and calcium imaging videos.

At the end of some recording sessions, mice received an intraperitoneal (i.p.) injection of either saline or amphetamine (3 mg/kg in saline) and were immediately placed back into the open field arena for an additional 45 min of behavior and calcium imaging following the same acquisition protocol. Amphetamine treatment always occurred on the last day of recording to prevent any effect on neuronal activity due to the long-lasting effects of amphetamine.

### Histology

After the behavioral experiments were completed, mice were deeply anesthetized with avertin (2,2,2-tribromoethanol 1.25%, 2-methyl-2-butanol 0.78%; 20 µl/g, i.p.; Sigma-Aldrich) and transcardially perfused with PBS followed by 4% paraformaldehyde in PBS. Brain were removed and postfixed overnight at 4 °C. Then, 40-µm coronal slices containing the striatum were cut with a vibratome (VT1000 S; Leica) and stored in PBS. Sections were washed for 10 min in PBS, incubated for 10 min with Hoechst 33258 (1:10,000 in PBS), and mounted on glass slides and coverslipped with Fluoromount. Slices were imaged using a microscope (V16; Zeiss) confirming for all mice the adequate localization of the lens in the dorsal striatum.

### Identification of behaviors

To identify the behaviors that mice displayed during open field explorations recorded through EthoVision XT14 (Noldus), we combined deep learning tools and clustering methods to generate a predictive model for labeling behaviors. First, the x-y coordinates of different mouse body parts (nose, neck, left ear, right ear, microendoscope camera, body center, tail start, and tail end) were identified using a DeepLabCut[18] deep neural network trained using 800 randomly selected and manually annotated frames taken from 40 different videos. The training regimen was set to the DeepLabCut default[18]. Any coordinate detected by DeepLabCut with a likelihood of less than 0.9 was removed from further analysis. For each video frame, the above body parts were used compute six features describing the mouse posture: the *body speed*, which was computed as the projection of the body center speed vector along the mouse body axis; the *head speed*, which was defined as the norm of the difference between the body center speed vector and the camera speed vector; the *movement angle*, which was calculated as the angle between the body center speed vector in the previous and subsequent frames; the *body length*, which was calculated as the sum of the distance between the neck and body center and the distance between the body center and tail start; the *neck elongation*, which was measured as the distance between the neck and body center; and the *head elevation*, which was calculated as the distance between the neck and the point defined by the orthogonal projection of the camera position along the vector orthogonal to the vector defined by ears positions. The temporal evolution of these six features was smoothed over 20 frames. Then, using a quarter of the data points, a nonlinear dimension reduction algorithm (t-distributed stochastic neighbor embedding, t-SNE) was applied to identify potent clusters in 3 dimensions, corresponding to the number of dimensions required to achieve ≥70% variance explained using principal component analysis. Ten replications of the t-SNE algorithm were computed for the same data. The clusters were then identified using a Gaussian

mixture model (GMM). Fifty replications of the GMM clustering with random initializations were calculated for each individual t-SNE replicate. The resulting 500 classifications of the frames were subsequently clustered using the Hamming distance to identify groups of frames that were consistently classified together by the tSNE and GMM methods. Clusters with less than 20 frames were removed. The remaining clusters were merged in ascending order using the Wasserstein distance (aka earth mover's distance) until a cutoff of 1 was reached. The resulting clusters were manually registered by visually inspecting the corresponding video frames and evaluating the distribution of cluster elements in the feature space as one of the following behaviors: *locomotion fast* (body speed greater than ~15 cm/s); *locomotion straight* (nonzero body speed, movement angle of ~0); *locomotion turn right* (nonzero body speed, nonzero positive movement angle); *locomotion turn left* (nonzero body speed, nonzero negative movement angle); *still turn right* (body speed of ~0 cm/s, nonzero positive movement angle); *still turn left* (body speed of ~0 cm/sec, nonzero negative movement angle); *head up* (body speed of ~0 cm/s, small neck elongation, high head elevation); *rearing* (body speed of ~0 cm/s, small body length, small neck elongation, high head elevation); *grooming* (speed of ~0 cm/s, nonzero camera speed, small body length, large movement angle variations); *locomotion sniffing* (nonzero body speed, large body length, high neck elongation); *still sniffing* (body speed of ~0 cm/s, large body length, high neck elongation); or *immobility* (body speed of ~0 cm/s, camera speed approximately of 0 cm/s). Finally, using these defined clusters and their distributions in the feature space, we evaluated for each frame its likelihood of belonging to each behavior cluster. The behavior was determined according to the highest likelihood. Any behavior episode of less than 100 ms (i.e., 4 frames) was removed. In cases in which some video frames were unlabeled, the first half of these series was attributed to the previous behavior, while the second half was attributed to the following behavior. The resulting identification of spontaneous mouse behavior in the open field exploration was systematically visually inspected to ensure proper classification.

## Validation of the behavior identification pipeline

In addition to the visual inspection, a pool of four annotators manually scored 2000 short video lasting 0.8–1.2 s created from a random selection of behavior episodes identified by the algorithm from all mice and from all recording sessions. All annotators had extensive experience in scoring mouse spontaneous behaviors. Video clips presentation was randomized between annotators. Annotation was performed using a custom MATLAB interface. To evaluate intra-annotator variability in scoring performance, all annotators were exposed twice to the same video clips. Annotators' classifications and predictions obtained from the behavior identification pipeline were compared to classes selected by annotator #1 (chosen as the annotator displaying the highest consistency between the two classifications of the same videos) and to a "majority classification", which was drawn by assigning for each observation the behavior class that received the higher number of votes from all annotators. Any observation that received less than 50% positive votes was discarded from the "majority classification".

To quantify the performance of behavior classification, we used the following metrics: accuracy, precision, recall, and F1 score. Accuracy was defined as the sum of true positives and true negatives over the total number of observations. Precision was defined as the number of true positives over the sum of true positives and false positives. Recall was defined as the number of true positives over the sum of true positives and false negatives. F1 score was defined as:

$$\text{F1 score} = 2 \times \frac{\text{precision} \times \text{recall}}{\text{precision} + \text{recall}} \tag{1}$$

All the above metrics were calculated for each individual behavior and average values combining all behaviors were calculated as the arithmetic mean of the per-class accuracies, precisions, recalls, and F1 scores with equal weights to each class.

## Calcium signal extraction, deconvolution and longitudinal cell registration

The calcium movies were preprocessed for spatial binning (downsampled by 4; OpenCV, Python) and subsequently motion-corrected and analyzed using CaImAn[19] to take advantage of the capabilities offered by the constrained nonnegative matrix factorization for endoscopic data (CNMF-E) algorithm to estimate and correct for background neuron somata[20]. The temporal CNMF-E components were manually curated to remove components with poor signal-to-noise ratios (peak-to-noise ratio of less than ~3), large baseline fluctuations, or inappropriate spatial footprints. The selected temporal calcium components were also deconvolved to estimate spike trains in the calcium measurements using MLspike[21]. To register cells across imaging sessions for the same animal, we used CellReg[30], which uses a probabilistic method to automatically register cells that are present in two or more recordings from the same mouse.

To control for longitudinal registration, two control lists of registered pairs of cells between pairs of sessions were generated: the first relied on a random shuffling of registered neurons in one of the two sessions, while the second relied on replacing all the neurons identified in one of the two sessions with their closest neighbor.

## Quantification of neuronal activation similarity and behavioral similarity

Two measures of neuronal similarity were used: the first compared the neuronal activation during each behavior over time during one recording session (or between recording sessions using pairs of longitudinally registered cells present in both sessions), while the second compared neuronal activation between different behaviors within a given recording session.

For the measure of the neuronal activation similarity for each behavior during one session (30 min long), we first split this session into two 15-min halves. For each time period, we calculated for each behavior the average value over time (15 min) of the deconvolved activity for each neuron, denoted as $\mathbf{X}_1$ and $\mathbf{X}_2$. The similarity was evaluated as:

$$-||\mathbf{X}_1/||\mathbf{X}_1|| - \mathbf{X}_2/||\mathbf{X}_2|| || \tag{2}$$

where $||\mathbf{X}||$ is the Euclidean norm of $\mathbf{X}$. As a control, the same similarity metric was computed by calculating $\mathbf{X}_1$ and $\mathbf{X}_2$ based on all possible partitions of 30 min into two periods using 5-min-long segments. We also calculated the neuronal similarity between odd and even frames by computing $\mathbf{X}_1$ and $\mathbf{X}_2$ in odd and even calcium frames. Finally, the neuronal activation similarity was also evaluated for each behavior by calculating $\mathbf{X}_1$ in one episode every two episodes and calculating $\mathbf{X}_2$ in the remaining episodes of each behavior. The spatial shuffle similarity was calculated as the mean of 10 random permutations of indices from $\mathbf{X}_2$.

In addition, the neuronal activation similarity between pairs of behaviors was calculated using the same formula as above. The average neuronal activity for each behavior was calculated over the duration of the entire session (30 min) except otherwise mentioned, and the similarity was computed for each pair of behaviors. The distance between behaviors (behavioral distance) was estimated for each pair of identified behaviors as the summation of the Wasserstein distances for each of the six features describing the mouse posture (body speed, head speed, movement angle, body length, neck elongation, and head elevation). The similarity between behaviors (behavioral similarity) was calculated as the opposite of the behavioral distance. Similar

results were obtained using similarity metrics based on the Wasserstein distance or Bhattacharyya distance. To evaluate the relationship between the neuronal activation similarity and the behavioral similarity, we used the Spearman correlation coefficient.

For all of the above experiments, if the sampling duration over which the average activity was computed was less than 5 s, the data were excluded from further analyses.

### Support vector machine decoding of behavior based on SPNs activity

To decode behaviors based on neuronal activity, we trained a set of binary support vector machine (SVM) classifiers for multiclass classification using the one-vs.-one strategy (scikit-learn Python package)[52]. Data were first split into a training set and a test set (80%/20% split). The training was performed using 5-fold cross validation to predict the detected behavior time series, with the deconvolved calcium activity convolved over a 500 ms square window. The outputs of the classifiers were combined, and the behavior with the highest number of votes was identified as the most likely behavior. The decoding accuracy was estimated using the test set as the fraction of time bins during which the predicted behavior corresponded to the observed behavior. The behavior reconstruction error was calculated using the behavior distance between the predicted and observed behaviors. Alternatively, to estimate the capability of SVM classifiers to separate one given behavior from any other behavior, which is referred in the text as one behavior vs. rest decoding, we calculated the accuracy as the simple matching coefficient between the observed and predicted behaviors (i.e., the true positive prediction for this behavior and the true negative prediction for this behavior).

The chance level for the decoding performance was obtained by training SVM classifiers on time-lagged data. Briefly, we flipped the behavior time series (the first element becomes the last and vice versa) and applied a cyclic permutation with a random time lag. This procedure destroys the relationships between the behavior and calcium activity time series but preserves the time correlations of the neural activity time series. For each recording session, 10 random time lags were used. For each random time lag, we trained a new set of SVM classifiers and evaluated its performance in predicting the original behavior using the original calcium time series. When plotted, the individual points for SVM decoding based on the shuffled data represent the average of the 10 random time lags.

For longitudinal predictions between pairs of recording sessions, the SVM classifiers were trained on one session using only neurons that were registered in these two sessions and the corresponding behavior time series. The decoding performance was evaluated using calcium events from the second session of cells registered in both sessions. If less than 40 neurons were identified in both sessions, the analysis was discarded.

### Detection of behavior-active neurons

To characterize the statistical significance of neuronal activation during identified behaviors, we employed a behavior information criterion that was calculated as the mutual information score between the calcium event occurrence and the mouse behavior. The behavior information for each cell was calculated using the following formula:

$$\mathrm{BI} = \sum_{i=1}^{N} p_i \frac{f_i}{f} \log 2 \frac{f_i}{f} \tag{3}$$

where $i$ is the behavior, $p_i$ is the fraction of time spent performing behavior $i$, $f_i$ is the average event frequency during behavior $i$, and $f$ is the overall average event frequency. We corrected this measure for sampling bias in the information measures by using shuffled distributions of the events. For each cell, we generated 1000 random permutations of the events and calculated a behavior information value for each permutation, thus generating 1000 random behavior information values to which the actual behavior information value was compared. We labeled a cell as a behavior-active cell if the behavior information value was more than 4 sigma from the shuffled distribution (significance of BI above 4)[26,27].

When this method was used in conjunction with SVM-based predictions, analyses were discarded if less than 40 neurons were labeled as behavior-active in a given session.

### Detection of behavior-silent (and behavior-inactive) neurons

To characterize cells that were consistently silent during a given behavior, we calculated for each neuron and each behavior the average event rate for all episodes of this behavior. We then evaluated for each cell and each behavior an activation occurrence value, which described how often this neuron was active during episodes of this behavior (event rate >0). For each behavior, a neuron was labeled as behavior-silent if its activation occurrence was less than 0.025. Alternatively, for each behavior, we labeled a cell as behavior-inactive if its average event rate during this behavior was less than 0.1 Hz.

When this method was used in conjunction with SVM-based predictions, analyses were discarded if less than 20 neurons were labeled as behavior-silent (or behavior-inactive) in a given session.

### Quantification and statistical report

Unless otherwise stated, the mean ± standard error of the mean (SEM) was used to report data. For all statistics, we used a linear mixed-effects model followed by analysis of variance to account for between-subject and within-subject effects in the case of incomplete design (exclusion criteria mentioned in the above sections). To compare the dSPN and iSPN groups, post hoc analyses were performed using permutation-based $t$ tests. For hypothesis testing, the significance was set to 0.05. Statistical analyses were performed in MATLAB (MathWorks). Animals were excluded prior to data acquisition if the imaging quality or focal plane were poor or after acquisition but before secondary analyses if movement artifacts were impossible to correct using CaImAn. The details of the statistical procedures and results are provided in Supplementary Table 1.

### Reporting summary

Further information on research design is available in the Nature Portfolio Reporting Summary linked to this article.

## Data availability

The data supporting the findings are available within the article and its Supplementary Materials and are available from the corresponding author upon request. Source data are provided with this paper.

## Code availability

DeepLabCut was used for processing the open field videos. CaImAn, CellReg, and MLspike were used for processing calcium imaging data. The custom codes used for this study are available on GitHub (https://github.com/deKerchove-Lab/Varin_2023_NatComm) with DOI (https://doi.org/10.5281/zenodo.8158538).

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

## Acknowledgements

We thank Dr. Philippe Faure and Dr. Clément Léna for their critical reading and their helpful insights on our manuscript. C.V. is a post-doctoral researcher of FRS-FNRS. A.C. is a research fellow of FRS-FNRS. P.B. is a research associate of the FRS-FNRS. A.K.E. is a research director of the FRS-FNRS and a WELBIO investigator. This research was supported by grants from FRS-FNRS (grants #23587797, #33659288, #33659296), WELBIO (#30256053), Fondation Simone et Pierre Clerdent (2018 Prize), and Fondation ULB to A.K.E., and grants from FRS-FNRS (grants #34793348, #35285205) to P.B. The funders had no role in study design, data collection and analysis, preparation of the manuscript, or decision to publish.

## Author contributions

C.V., P.B., and A.K.E. designed the study. A.C and D.H. deployed in vivo calcium imaging protocols and performed experiments. C.V., A.C., and A.K.E. conceptualized the analyses. C.V. and A.C. established and conducted calcium signal extraction and behaviors identification. C.V. performed the other analyses. A.K.E. supervised the project. C.V., A.C., and A.K.E. wrote the manuscript. All authors discussed the manuscript.

## Competing interests

The authors declare no competing interests.
