## [Peer Review File · Nature Communications]

The respective activation and silencing of striatal direct and indirect pathway neurons support behavior encodingREVIEWER COMMENTS

Reviewer #1 (Remarks to the Author):

Varin et al., developed a very large experimental and analytical project, in which they recorded the activity of direct- and indirect- pathway neurons in the dorsal striatum of freely-behaving mice using miniscopes. The recordings are behaviorally annotated utilizing a custom-built pipeline for annotating video-streams (based on deep-lab-cut). Based on the analysis of the activity of direct- and indirect- pathway activation in the context of behavioral annotation, the authors propose a revision of the 'selection-suppression' model, which they call the 'adaptive selection suppression model'.

The study appears to be a deep analysis, and has the potential to provide an important update to the current leading models regarding the striatal computations underlying action selection.

However, a few major issues limit the capacity to properly evaluate the work and support it towards publication. These are defined below.

I will be happy to provide a more detailed and full review of the manuscript, in support of publication, following a significant revision of the work, addressing the comments below.

I should note that the majority of the analysis included in later figures leave the sense that this is a well thought-out and carefully implemented analysis, supporting a novel revision of the current working hypothesis regarding striatal action selection. This provides me with the confidence that an initial revision, which will improve clarity and ensure the validity of the behavioral annotation is worthwhile.

Major point (#1): This manuscript is quite difficult to read. A first step towards improving the clarity of the manuscript would be to make the title and abstract focus on the bottom line of the revised model that is proposed, in-line with the flow of the introduction to the manuscript.

Secondly, the behavioral annotation is crucial for the rest of the manuscript. It is important to explain the strategy and provide evidence for the quality of annotation, and not leave this to the supplementary data and methods alone.

From a practical perspective, it would ease the review process tremendously if the authors would include the legends with the figures, and also (albeit less crucial), if they would embed the figures in the relevant positions in the manuscript.

Also, making the statistical approaches more accessible (in the legends or the results) would also be very helpful. For example, viewing the distribution of the individual data points in Figure 1F (the first point about which a statement of significance is made, it is very hard to be convinced that there is a statistically significant difference here. To address this, one goes to the legend, and does not find an answer, and only upon going to the stats table (in an additional file), does one find any description of

the stats (permutation based t-test), but no additional information is provided (p-value etc'). This does not help the reviewer gain confidence in the project.

Major point (#2): The work is largely based on the development of a behavioral annotation pipeline, which is schematized in Figure S1. This appears to be an elegant pipeline, based on Deeplabcut for labeling, followed by feature extraction and clustering, and then manual labeling of clusters to define the behaviors specified. As this is a center-piece of the project, upon which hinges so much of the annotation of the physiological signals, it would be important to provide further explanation of the methodology within the body of the text, as well as validation of the annotation with regard to an unbiased human observer. What are the efficiency and reliability of the behavioral annotation? Without this data, it is very difficult to evaluate the rest of the manuscript.

Major point (#3): The analysis involves a deconvolution of the calcium imaging to 'spiking' events. Is this essential for the analysis? Any manipulation on the data that is not essential could introduce biases and mis-interpretations. Since the authors perform calcium imaging, rather than electrophysiological recordings, why not remain within the realm of the data measured?

More importantly, the authors describe the activity (extrapolated from the deconvoluted calcium signal) that corresponds to the performance of an action (presumably throughout the event). Is this the most relevant metric? It would be very helpful if the authors could provide concrete examples of calcium activity associated with specific behavioral events. Does the activity precede the event? Is it maintained throughout the event? Perhaps the relevant striatal computation is whether to engage in an action, and this is performed before the action is initiated? Parsing the activity of neurons to different time-points with relation to the performance of an action may be crucial for proper evaluation of the model the authors are proposing - neurons may be dedicated to action initiation, others (or non at all) to maintenance of an action, and yet others to action termination. This would require a significant revision of the analysis.

Reviewer #2 (Remarks to the Author):

The manuscript by Varin and colleagues investigates how striatal direct and indirect pathway neurons encode different aspects of spontaneous open field behavior using 1-photon calcium imaging (i.e., miniscope). The authors use supervised learning techniques to uncover the encoding properties of striatal neurons during behavior. Specifically, they see that direct pathway neurons encode various behaviors through their activation, whereas indirect pathway neurons encode behaviors by silencing their activity. The authors build upon previous work in the field (Markowitz, Klaus/Martin, Barbara etc...) to provide an updated model of striatal function and put their work in context with previous studies.

This work is timely and would be of broad interest to readers. Overall, we feel positive about this study; however, several points need additional clarification as well as greater justification for the experimental approach.

Major comments:

- 1) More detail and clarification is needed on the methods underlying the behavior detection pipeline. Specifically, 1) are behaviors manually or automatically clustered? The authors mention t-SNE, GMM, and Hamming distance clustering to categorize behaviors and then report manually inspecting the clusters: "The resulting clusters were manually registered by visually inspecting the corresponding video frames and evaluating the distribution of cluster elements in the feature space as one of the following behaviors...". Thus it is unclear how much is manual vs. automated. What is the justification for this hybrid process? How different would behavioral events look without manual inspection? Please elaborate. 2) Why is t-SNE being used in a quantitative manner? As it's typically used for only visualization of high-dimensional data, we found it odd that it was the primary clustering algorithm. Is the tSNE done on data post PCA (to reduce dimensionality) or is it being used to reduce dimensionality? The latter is generally not the right way to use it.
- 2) The algorithm seems to embrace randomness at certain steps and then take a winner takes all approach. Why? How much disagreement is there among the "votes" in the 2 random steps?
- 3) How was the pipeline benchmarked?
- 4) Importantly, how is spontaneous behavior aligned to calcium imaging data? The authors go into detail about the pre-processing for each modality (behavior and imaging, Supp Fig 1) but specifics for how the two are aligned are not mentioned. Please clarify or add to Supp Fig 1.
- 5) For your neuronal similarity measure, what is the justification to use $1 - \frac{\|X_1 - X_2\|}{\|X_1\| + \|X_2\|}$, where $\|X\|$ is the Euclidean norm of X ? Why not just use something more conventional and take the dot product between the two vectors? If I read the equation correctly, then two vectors at 90 degrees to each other (i.e. orthogonal) give a value here of $1 - \frac{\sqrt{2}}{2} \sim 0.707$??
- 6) For training of the SVM classifiers, the manuscript states: "The training was performed using 5-fold cross validation to predict the detected behavior time series, with the deconvolved calcium activity convolved over a 500 ms square window". From this description it is unclear if there was a testing vs. training data set. Was any data withheld to test the final SVMs, as it is common practice?
- 7) From the methods, it is unclear what side of the brain the recordings were made. Were both right and left hemispheres sampled? Given that the basal ganglia and striatum are known to be important for lateralized movements/tasks it would be interesting to analyze or at least comment on the laterality of spontaneous movements aligned with calcium activity as this could change interpretation of results.
- 8) Could you please show us the distribution of behavior information indexes for all cells? What fraction of cells pass this?

Minor comments:

- 1) Please state the atlas your anatomical coordinates correspond to—i.e. Allen Institute or Franklin/Paxinos?
- 2) Your models in Supp Fig 10 are very nice and a great conclusion to the manuscript.

Reviewer #3 (Remarks to the Author):

In this manuscript, Dr. Vartin and colleagues address a long-standing question of how D1 and D2 containing MSN activity contribute to behavior. Using calcium imaging in freely moving mice engaged in naturalistic behavior, they first classified discrete clusters of behavior, then characterized activity of these different populations during behavior, and used machine learning to determine how firing rates of decoded behavior. They report that while behavior can be equally well decoded from both D1 and D2 MSNs, D1 MSNs are more activated by behaviors than D2 MSNs, and less variable over time. By contrast, a larger portion of D2 MSNs are behaviorally silent. Correspondingly, behaviorally activated D1 MSNs do a better job of decoding behavior than behaviorally activated D2 MSNs. In contrast, for D2 MSNs, in several behaviors, behaviorally silent neurons outperformed non-silent neurons, a pattern not frequently seen in D1 MSNs. From this set of data, the authors propose an interesting new model of D1 and D2 MSN cooperative activity to shape behavior in which D1 MSNs encode a variety of similar behaviors, and selective D2 MSN silence for a particular behavior along with concomitant activity to similar behaviors promoting behavioral specificity.

Overall, I find the question very interesting, the dataset well suited to address it, and the hypothesized model novel and exciting. However, I am concerned that the current analyses and interpretations do not fully support the authors' conclusions.

1. The analyses of firing rate are underdeveloped. The main metric used for several figures is activation similarity, essentially whether the average firing rate during a behavior is the same each time it happens. This metric rests on the premise that absolute firing rate per se (as opposed to its dynamic movement up or down) is the unit conveys information. Not surprisingly, this metric isn't very high across time (on average 0.4) and is also fairly similar across several different behaviors (figure 2e). It is unclear why the authors picked this particular metric (which involves a fair bit of processing of the data) over other, more direct ways of examining the data. For instance, if the authors wanted to make the point that D1 MSNs have more consistent response patterns than D2, they could have looked at a measure of variability (e.g., $1/CV$) of behaviorally induced firing rates.
2. Similarly, I was puzzled by the analysis choices for determining what constitutes behaviorally activated and silenced activity. A straightforward way of calculating this would be to ask whether behavior associated firing rates were statistically higher or lower than those occurring by chance (i.e. with a shuffle). Instead, the authors chose mutual information as the definition behavioral activation

and 1 of 2 arbitrary thresholds for behavioral silencing. For the former, I couldn't find where the authors show that this will only result in neurons that are behaviorally activated. It seems to me that cells that consistently decrease their activity in conjunction with a particular behavior will have high mutual information with that behavior. For the latter, that definition is going to over represent neurons with generally low firing rates. At the extreme, a cell that never fires will be behaviorally silent to all behaviors. Indeed, D2 MSNs have lower firing rate and more behaviorally silent cells. This problem is rendered greater by the fact that the calcium imaging is capturing firing rates that are 10-fold lower than those recorded in the same neurons with electrophysiology, suggesting that much low frequency activity of these neurons is going unrecorded.

3. Several of the decoding results seem like they could be explained by how many neurons were used to decode rather than the identify of the neurons. For example, there are more "behavior active" D1 MSNs than there are D2 MSNs (4b), thus potentially explaining the results in 4c. Similarly, there are more "behavior silent" D2 MSNs than D1. Were matched numbers of cells used for decoding across all conditions?

4. It isn't clear why figures 3 and 4 use one measure of decoding, while figure 5 uses a different measure. This makes it difficult to compare the performance of behaviorally active vs behaviorally silent.

5. The authors argue that D1 MSNs neural coding is biased towards activation while D2 MSNs are biased towards silencing. This may be true (with the all the caveats listed above), but we are talking about small differences. Forty percent of the D1 MSNs are not "behaviorally activated" and those neurons still perform above chance levels at decoding (4c). Similarly, for the most part, the small proportion of "behaviorally silent" D1 MSNs neurons don't perform at decoding any worse than their non-silent counterparts (and again seem to perform above chance levels). It was unclear how the authors accounted for these findings which seem counter to their model.

6. The authors propose an interesting model (described above). If I have understood their model correctly, one would predict that for a given behavior A, average D2 MSN activity should be statistically lower than average for A, but statistically higher for all behaviors similar to A. The authors should test that prediction with the data they have.

7. In extended figure 4, please show the effect of amphetamine on firing rates for D1 and D2 MSNs.

Minor

1. Why don't the values on lines 184-185 match those in figure 3b?

2. Overall, the manuscript is very well-written. There are some typos in the figure legend (line 630, 633 and 793 should read 'blue')

3. Does the color coding for the behavior have numerical meaning in terms of distance? If so, it should be labeled accordingly.

4. Some of the statistical legends were unclear. For example, in extended figure 4d, the D2 MSN and shuffle reconstruction bars post amphetamine have nearly identical values but are listed as having ### $p < 0.001$. Is this correct? But there are also lines with astrices that have associated different values...

Revision of manuscript NCOMMS-22-36538 entitled “The respective activation and silencing of striatal direct and indirect pathway neurons support behavior encoding”.

REVIEWER COMMENTS

Reviewer #1 (Remarks to the Author):

Varin et al., developed a very large experimental and analytical project, in which they recorded the activity of direct- and indirect- pathway neurons in the dorsal striatum of freely-behaving mice using miniscopes. The recordings are behaviorally annotated utilizing a custom-built pipeline for annotating video-streams (based on deep-lab-cut). Based on the analysis of the activity of direct- and indirect- pathway activation in the context of behavioral annotation, the authors propose a revision of the 'selection-suppression' model, which they call the 'adaptive selection suppression model'.

The study appears to be a deep analysis, and has the potential to provide an important update to the current leading models regarding the striatal computations underlying action selection.

However, a few major issues limit the capacity to properly evaluate the work and support it towards publication. These are defined below.

I will be happy to provide a more detailed and full review of the manuscript, in support of publication, following a significant revision of the work, addressing the comments below.

I should note that the majority of the analysis included in later figures leave the sense that this is a well thought-out and carefully implemented analysis, supporting a novel revision of the current working hypothesis regarding striatal action selection. This provides me with the confidence that an initial revision, which will improve clarity and ensure the validity of the behavioral annotation is worthwhile.

Major point (#1): This manuscript is quite difficult to read. A first step towards improving the clarity of the manuscript would be to make the title and abstract focus on the bottom line of the revised model that is proposed, in-line with the flow of the introduction to the manuscript.

According to Reviewer request, we modified the title and the abstract of the manuscript.

Secondly, the behavioral annotation is crucial for the rest of the manuscript. It is important to explain the strategy and provide evidence for the quality of annotation, and not leave this to the supplementary data and methods alone.

Please find below, in Major point #2, a response to this query.

From a practical perspective, it would ease the review process tremendously if the authors would include the legends with the figures, and also (albeit less crucial), if they would embed the figures in the relevant positions in the manuscript.

We modified the organization of figures and captions in our revised manuscript file.

Also, making the statistical approaches more accessible (in the legends or the results) would also be very helpful. For example, viewing the distribution of the individual data points in Figure 1F (the

first point about which a statement of significance is made, it is very hard to be convinced that there is a statistically significant difference here. To address this, one goes to the legend, and does not find an answer, and only upon going to the stats table (in an additional file), does one find any description of the stats (permutation based t-test), but no additional information is provided (p-value etc'). This does not help the reviewer gain confidence in the project.

Following Reviewer's advice, we updated figure captions to provide information about the statistical tests used.

Major point (#2): The work is largely based on the development of a behavioral annotation pipeline, which is schematized in Figure S1. This appears to be an elegant pipeline, based on Deeplabcut for labeling, followed by feature extraction and clustering, and then manual labeline of clusters to define the behaviors specified. As this is a center-piece of the project, upon which hinges so much of the annotation of the physiological signals, it would be important to provide further explanation of the methodology within the body of the text, as well as validation of the annotation with regard to an unbiased human observer. What are the efficiency and reliability of the behavioral annotation? Without this data, it is very difficult to evaluate the rest of the manuscript.

We agree with Reviewer #1 that this critical information was missing in our original manuscript. According to Reviewer's request we added a description of our behavior segmentation pipeline in the result section. We also added in our revised manuscript a quantification of the performance of the behavior annotation pipeline in comparison with human annotators (Supplementary Fig. 2h-k and corresponding results).

Major point (#3): The analysis involves a deconvolution of the calcium imaging to 'spiking' events. Is this essential for the analysis? Any manipulation on the data that is not essential could introduce biases and mis-interpretations. Since the authors perform calcium imaging, rather than electrophysiological recordings, why not remain within the realm of the data measured?

Signals from calcium-sensitive fluorescent probes are an indirect readout of neuron spiking activity. Fluorescent reporters have kinetics (rise time ~ 200 ms; decay time ~ 1 s) that are slower than that of the underlying spikes and act as slow integrator over spiking activity. These kinetics are in a range similar to the duration of behaviors we identified (average episodes duration: 0.5-0.8 s; Supplementary Fig. 2m). As a consequence, the correlations between calcium signal and behaviors are likely strongly attenuated. Calcium signal from a behavior-specific neuron could still be high when the animal engages in a different behavior.

This deconvolution step was critical for our analyses. Indeed, even if the same conclusions can be drawn from the evaluation of the average activity per behaviors using fluorescence signal (Supplementary Fig. 3i) or deconvolved activity (Fig. 1g), this was not the case for subsequent downstream analyses. When neuronal activation similarity is computed from the fluorescence signal (Supplementary Fig. 4g), values obtained are often not significantly different than those obtained by chance, as evaluated by a random spatial shuffle. This precludes any decisive conclusions about the significant differences between dSPNs and iSPNs that can be observed. These effects are most likely due to the long-lasting calcium transient decay and the temporal offset between the underlying spiking train (or its deconvolved estimate) and the peak in calcium fluorescence. Because of these considerations and observations, we decided to use deconvolved signal.

More importantly, the authors describe the activity (extrapolated from the deconvoluted calcium signal) that corresponds to the performance of an action (presumably throughout the event). Is this the most relevant metric? It would be very helpful if the authors could provide concrete examples of calcium activity associated with specific behavioral events. Does the activity precede the event? Is it maintained throughout the event? Perhaps the relevant striatal computation is whether to engage in an action, and this is performed before the action is initiated? Parsing the activity of neurons to different time-points with relation to the performance of an action may be crucial for proper evaluation of the model the authors are proposing - neurons may be dedicated to action initiation, others (or non at all) to maintenance of an action, and yet others to action termination. This would require a significant revision of the analysis.

According to Reviewer's query, we now provide examples of neuron responses around behavior onsets or offsets (Supplementary Fig. 7d) as well as averaged responses of behavior-active and behavior-silent dSPNs and iSPNs around behaviors onsets (Supplementary Fig. 7e and Supplementary Fig. 8a). In our recordings, we could not find any evidence of a temporal organization of neuronal activation during episodes or around episodes onsets or offsets. Activations of behavior-active neurons and decreases in activity of behavior-silent neurons are aligned to behaviors onsets and appear consistent throughout the duration of episodes. Our observations are in line with those harvested under similar conditions (e.g. Klaus et al, 2017, Neuron; Weglage et al., 2012, Cell Rep; Markowitz et al., 2018, Cell).

We agree with Reviewer #1 that the dorsal striatum definitely plays a pivotal role in goal directed behaviors and action planning with, for the execution of some elaborated behaviors, neurons encoding different aspects of behavioral sequence structure: some neurons encoding each or particular elements, some encoding the start and/or the completion of the sequence (e.g. Geddes et al., 2018, Cell). However, these characterizations rely on the execution (and repetition) by highly trained animals of complex goal-directed behaviors that take many seconds to be completed.

On the other hand, our recordings were performed in mice freely exploring and open field at their own pace without any kind of incentive to engage in any particular behavior. Mice shift from one behavior to the other roughly once every 0.5-0.8 s. Despite the fact that the alteration between spontaneous behaviors is not random (Supplementary Fig. 2n), we postulate that few to no long-term action sequence planning is engaged during spontaneous open field exploration.

However, it would be interesting, in future experiments, to test if what we observed during spontaneous self paced behaviors holds true during the execution of goal-directed operant actions.

Reviewer #2 (Remarks to the Author):

The manuscript by Varin and colleagues investigates how striatal direct and indirect pathway neurons encode different aspects of spontaneous open field behavior using 1-photon calcium imaging (i.e., miniscope). The authors use supervised learning techniques to uncover the encoding properties of striatal neurons during behavior. Specifically, they see that direct pathway neurons encode various behaviors through their activation, whereas indirect pathway neurons encode behaviors by silencing their activity. The authors build upon previous work in the field (Markowitz, Klaus/Martin, Barbara etc...) to provide an updated model of striatal function and put their work in context with previous studies. This work is timely and would be of broad interest to readers. Overall, we feel positive about this study; however, several points need additional clarification as well as greater justification for the experimental approach.

Major comments:

1) More detail and clarification is needed on the methods underlying the behavior detection pipeline. Specifically, 1) are behaviors manually or automatically clustered? The authors mention t-SNE, GMM, and Hamming distance clustering to categorize behaviors and then report manually inspecting the clusters: “The resulting clusters were manually registered by visually inspecting the corresponding video frames and evaluating the distribution of cluster elements in the feature space as one of the following behaviors...”. Thus it is unclear how much is manual vs. automated. What is the justification for this hybrid process? How different would behavioral events look without manual inspection? Please elaborate.

We understand Reviewer’s comment arguing our behavior annotation pipeline was not explained clearly enough. As a consequence we added a more detailed description of the behavior annotation procedure in the beginning of the result section.

Briefly, the workflow for behavior annotation consisted of two parts: an unsupervised detection of behavior clusters, followed by a manual registration of these clusters into one of the 12 behaviors capturing the behavioral repertoire expressed by mice in the open field.

The first unsupervised step tends to over-segment what would be defined as the same behavior (Figure 1 below). For example we often observed a separation within bouts of locomotion with turns between different parts of the same action: the first few hundreds milliseconds during which movement angle remains small (around 10-20°) belonged to one cluster and a second part when movement angle increases (up to 40°) belonged to a separated cluster. We tried to modify our algorithm to merge these clusters defining different portions of the same behavior, but it also resulted in the merging of actually different behaviors (e.g. grooming and still with turns). Moreover, these clusters already contained more elaborated actions than the syllables identified by the Datta’s group (Wiltschko et al., 2015, Neuron; Markowitz et al., 2018, Cell).

To solve this issue, we decided to manually register clusters corresponding to different stages of the same behavior to maintain a proper separation between different behaviors. This idea to manually register cluster into a defined list of consistent behaviors also aimed at being able to directly compare the same behaviors list and its neural correlates between different animals. Keeping the final list of behaviors relatively short (12 behaviors compared to the ~25-30 clusters before registration or the > 40 syllables identified by Datta’s group) was also important for subsequent analyses as it enabled us to gather a sufficiently large amount of data points belonging to each individual behavior.

Figure 1. Behavior annotation before and after manual registration of behaviors.
a-b, Representative example of consistent behavior clusters (#1 to #31) identified after multiple iterations of tSNE followed by GMM clustering (**a**), and their subsequent manual registration into one of the 12 behaviors (**b**).
c-d, Behaviors identification using either behavior clusters before (**c**) or after (**d**) manual registration. In **c**, clusters are organized and colored according to their subsequent manual registration
e-f, Average duration of episodes detected using behavior clusters before manual registration (**e**) or after manual registration of behaviors (**f**). In **e**, dotted horizontal lines correspond to average values in **f**.

2) Why is t-SNE being used in a quantitative manner? As it's typically used for only visualization of high-dimensional data, we found it odd that it was the primary clustering algorithm. Is the tSNE done on data post PCA (to reduce dimensionality) or is it being used to reduce dimensionality? The latter is generally not the right way to use it.

We understand Reviewer's comment as running clustering methods after non-linear dimension reduction is somewhat controversial. Many of the concerns focus on the fact that t-SNE does not completely preserve density and that it can create tears within actual clusters.

We choose t-SNE over PCA for dimension reduction because PCA does not retain non-linear variance whereas t-SNE retains local variance. It provides improved ability to delineate high dimensional data in a low dimension space over linear dimension reduction methods. The number of dimensions for t-SNE was chosen according to the number of dimensions required to obtain at least 70% variance explained using PCA. Because t-SNE destroys global distances, we then used a non-distance based clustering method to separate potent clusters. As detailed in response to the following query, we were

aware of the non-deterministic nature of both t-SNE and GMM, and we ran multiple iterations of t-SNE followed by GMM to extract only data points that consistently end up clustered together after each iteration (Hamming clustering on the outputs of all t-SNE + GMM iterations). Then using the distribution of clusters in the feature space (body speed, movement angle, ...), we estimated the likelihood of each data point of belonging to one of the other behaviors.

Similar approaches have been employed for the unsupervised segmentation of animals' behaviors (e.g., Berman et al., 2016, PNAS; Dolensek et al., 2020, Science).

2) The algorithm seems to embrace randomness at certain steps and then take a winner takes all approach. Why? How much disagreement is there among the “votes” in the 2 random steps?

Indeed, because both t-SNE and GMM clustering algorithms are non-deterministic (and thus highly sensitive to initial conditions), we decided to run them multiple times with random seeds to ensure the consistency and the reproducibility of final clusters by selecting only behavioral features data points that consistently end up clustered together after each iteration of t-SNE and GMM. We estimate the agreement between the partitions resulting from two different iterations of t-SNE followed by GMM clustering at 0.8813 ± 0.0024 (Rand Index). These disagreements are most likely due to randomized seeds used for the initiation of both the t-SNE and the GMM clustering.

3) How was the pipeline benchmarked?

As also pointed by Reviewer #1, this key information was missing in our original manuscript. We now provide a quantification of the performance of the behavior annotation pipeline in comparison with human annotators (Supplementary Fig. 2h-k and corresponding results).

4) Importantly, how is spontaneous behavior aligned to calcium imaging data? The authors go into detail about the pre-processing for each modality (behavior and imaging, Supp Fig 1) but specifics for how the two are aligned are not mentioned. Please clarify or add to Supp Fig 1.

We agree with Reviewer #2 that this information explaining how we made sure both calcium and behavior videos are properly aligned were missing in our original submission. Quickly, during recording sessions, the TTL signal emitted by EthoVision to trigger calcium imaging was recorded by the Inscopix DAQ box and the ON/OFF status of the microscope emitted by the Inscopix DAQ box was recorded by EthoVision. The comparison between both signals enabled us to correct delays ($< \sim 100$ ms) between behavior and calcium imaging videos and to correct differences in acquisition rates ($< \sim 0.01$ fps) between the camera recording behavior and the microendoscope.

This information was added in the methods in our revised manuscript.

5) For your neuronal similarity measure, what is the justification to use $1 - \frac{\|X1 - X2\|}{\|X1\| + \|X2\|}$, where $\|X\|$ is the Euclidean norm of X ? Why not just use something more conventional and take the dot product between the two vectors? If I read the equation correctly, then two vectors at 90 degrees to each other (i.e. orthogonal) give a value here of $1 - \frac{\sqrt{2}}{2} \sim -0.4$??

We decided to use a similarity measure based on the Euclidean distance between a pair of vectors to be as close as possible to the measure used by Klaus and colleagues in their article (Klaus et al., 2017, Neuron). This enabled us to directly compare our observations to the ones they harvested for the comparison of neuronal activation between pairs of behaviors (Fig. 3; Figure 4 in Klaus et al., 2017) and importantly explain why we observed something they did not observe (Supplementary Fig. 6).

In addition, we wanted to consistently use the same measure to compare neuronal activity (*i*) during the same behavior within the same recording session (Fig. 2), (*ii*) between pairs of behaviors during the same recording session (Fig. 3), and (*iii*) during the same behavior for longitudinally registered neurons between pairs of recording sessions (Fig. 7).

However, the use of the “1-distance” to transform the Euclidean distance into a similarity measure was indeed an odd choice as the Euclidean distance we computed can be larger than 1. We thus removed this “1” and changed the similarity measure into the opposite of the Euclidean norm of the difference between the neuronal activity vectors.

Moreover, following Reviewer’s suggestion, we illustrate in the revised manuscript a measure of neuronal activation similarity using the dot product (Supplementary Fig. 4f). Results are qualitatively similar to those obtained with the Euclidean similarity.

6) For training of the SVM classifiers, the manuscript states: “The training was performed using 5-fold cross validation to predict the detected behavior time series, with the deconvolved calcium activity convolved over a 500 ms square window”. From this description it is unclear if there was a testing vs. training data set. Was any data withheld to test the final SVMs, as it is common practice?

We thank Reviewer #2 for pointing this key methodological step we forgot to include when we described SVM-related methods. Methods are corrected accordingly in the revised manuscript.

7) From the methods, it is unclear what side of the brain the recordings were made. Were both right and left hemispheres sampled? Given that the basal ganglia and striatum are known to be important for lateralized movements/tasks it would be interesting to analyze or at least comment on the laterality of spontaneous movements aligned with calcium activity as this could change interpretation of results.

Calcium imaging recordings were consistently made in the left dorsal striatum. This missing information was added in the corresponding method section (“Viral injections and chronic lens implantation”).

As pointed by Reviewer #2, we did not make any comment regarding the lateralization in the dorsal striatum. We agree that this is an important point to address. In our original manuscript, it was already possible to appreciate (Supplementary Fig. 7f, formerly ED Fig 6c) the higher proportion of behavior-active dSPNs during contralateral turns (right turns) than during ipsilateral turns (left turns) with or without locomotion. For iSPNs we found a larger proportion of behavior-active neurons during contralateral right turns with locomotion. The activation profiles of these behavior-active neurons around their respective behaviors onsets are qualitatively similar between left and right turns (Supplementary Fig. 7f). We added some text in the result section to show that we indeed observed a lateralization in the coding of contralateral and ipsilateral turns.

8) Could you please show us the distribution of behavior information indexes for all cells? What fraction of cells pass this?

We added the representation of the distribution of behavior information in Supplementary Fig. 7c. Moreover we would like to reinforce the idea that the absolute values of behavior information are not the measure over which we defined cells as behavior-active. Indeed mutual information is biased toward low firing neurons (Supplementary Fig. 7a-b). To overcome this issue, we compared for each cell the actual value of mutual information to the distribution of mutual information calculated using shuffled distributions of the events to calculate a measure of significance of behavior information.

Irrespective of the absolute value of behavior information, cells were classified as behavior-active when the significance of behavior information exceeds 4. The fraction of cells exceeding this criterion and thus classified as behavior-active is displayed in Figure 4b (right panel)

Minor comments:

1) Please state the atlas your anatomical coordinates correspond to—i.e. Allen Institute or Franklin/Paxinos?

We added this information into the corresponding method section

2) Your models in Supp Fig 10 are very nice and a great conclusion to the manuscript.

Thank you

Reviewer #3 (Remarks to the Author):

In this manuscript, Dr. Varin and colleagues address a long-standing question of how D1 and D2 containing MSN activity contribute to behavior. Using calcium imaging in freely moving mice engaged in naturalistic behavior, they first classified discrete clusters of behavior, then characterized activity of these different populations during behavior, and used machine learning to determine how firing rates of decoded behavior. They report that while behavior can be equally well decoded from both D1 and D2 MSNs, D1 MSNs are more activated by behaviors than D2 MSNs, and less variable over time. By contrast, a larger portion of D2 MSNs are behaviorally silent. Correspondingly, behaviorally activated D1 MSNs do a better job of decoding behavior than behaviorally activated D2 MSNs. In contrast, for D2 MSNs, in several behaviors, behaviorally silent neurons outperformed non-silent neurons, a pattern not frequently seen in D1 MSNs. From this set of data, the authors propose an interesting new model of D1 and D2 MSN cooperative activity to shape behavior in which D1 MSNs encode a variety of similar behaviors, and selective D2 MSN silence for a particular behavior along with concomitant activity to similar behaviors promoting behavioral specificity.

Overall, I find the question very interesting, the dataset well suited to address it, and the hypothesized model novel and exciting. However, I am concerned that the current analyses and interpretations do not fully support the authors' conclusions.

1. The analyses of firing rate are underdeveloped. The main metric used for several figures is activation similarity, essentially whether the average firing rate during a behavior is the same each time it happens. This metric rests on the premise that absolute firing rate per se (as opposed to its dynamic movement up or down) is the unit conveys information. Not surprisingly, this metric isn't very high across time (on average 0.4) and is also fairly similar across several different behaviors (figure 2e). It is unclear why the authors picked this particular metric (which involves a fair bit of processing of the data) over other, more direct ways of examining the data. For instance, if the authors wanted to make the point that D1 MSNs have more consistent response patterns than D2, they could have looked at a measure of variability (e.g., $1/CV$) of behaviorally induced firing rates.

There are two reasons for which we used this measure. First, we wanted to consistently use the same metric to compare neuronal activity (i) during the same behavior within the same recording session

(Fig. 2), (ii) between pairs of behaviors during the same recording session (Fig. 3), and (iii) during the same behavior for longitudinally registered neurons between pairs of recording sessions (Fig. 7). Doing so (especially for the comparison between behaviors) requires comparing pairs of neuronal activity vectors. Secondly, we wanted to use a similarity measure as close as possible to the one used by Klaus and collaborators (Klaus et al., 2017, Neuron) in their article highlighting a continuous representation of behaviors within the neural space. They compared neuronal activation between behaviors using the norm of the difference between neuronal activity vectors (Euclidean distance). We thus used a similar Euclidean measure instead of any of the numerous other distance metric to evaluate a distance between vectors (e.g. dot product, cosine similarity, Manhattan distance, Hamming distance, ...).

Moreover the use of this similarity measure was useful to provide additional evidence that differences in neural activation between dSPNs and iSPNs was not something that resulted from a differential integration through time of some internal drives that accumulate or decrease during the 30-min of open field exploration (Supplementary Fig. 4c).

Despite not stated in our initial submission, we tested other similarity measures between pairs of vectors and they produced qualitatively similar results. Following advice from Reviewer #2 we report in our revised manuscript the results of neuronal activation similarity using the dot product (Supplementary Fig. 4f). The same differences between dSPNs and iSPNs are observed with this similarity measure.

Notwithstanding the above points, we thank Reviewer #3 for suggesting the use of $1/CV$ as a measure of neuronal variability during each behavior. We report this measure in our revised manuscript (Fig. 2c): $1/CV$ is significantly higher in dSPNs than in iSPNs for most of the behaviors. It's an important addition to our manuscript as it enables us to reinforce the idea that for each behavior, the same dSPNs are more consistently activated, whereas iSPN activation is more variable using a complementary measure that constitutes a key additional control.

2. Similarly, I was puzzled by the analysis choices for determining what constitutes behaviorally activated and silenced activity. A straightforward way of calculating this would be to ask whether behavior associated firing rates were statistically higher or lower than those occurring by chance (i.e. with a shuffle). Instead, the authors chose mutual information as the definition behavioral activation and 1 of 2 arbitrary thresholds for behavioral silencing. For the former, I couldn't find where the authors show that this will only result in neurons that are behaviorally activated. It seems to me that cells that consistently decrease their activity in conjunction with a particular behavior will have high mutual information with that behavior. For the latter, that definition is going to over represent neurons with generally low firing rates. At the extreme, a cell that never fires will be behaviorally silent to all behaviors. Indeed, D2 MSNs have lower firing rate and more behaviorally silent cells. This problem is rendered greater by the fact that the calcium imaging is capturing firing rates that are 10-fold lower than those recorded in the same neurons with electrophysiology, suggesting that much low frequency activity of these neurons is going unrecorded.

In information theory, mutual information qualifies the amount of information that is shared between two variables (in our case, the distribution of behaviors and the calcium activity in one neuron). Intuitively, it quantifies how much knowing one variable predicts the other variable. When the two variables are independent, the mutual information is equal to zero. The other way around, when the two variables are deterministic functions of each other, the mutual information is maximal. In the figure below are illustrated some examples representing how mutual information changes depending on neuronal activity.

Figure 2. Relationship between mutual information and neuronal activity. Examples of mutual information (MI) calculation as a function of the average activity in a situation for which all states last the same amount of time (top left panel). MI is minimal when the activity is equal during all states and is maximal when the activity occurs during only one state.

Mutual information is maximal when activity is limited to a single state, and is zero when activity is equal during all states. For the special case in which activity is zero during one single state (state A in the example illustrated; bottom right panel) and similar during all the other states, mutual information remains low because, according the formula, in the summation over states, activity during state A contributes to zero to mutual information, and the other states, because their relative activity (f_i/f) is close to one, slightly increase mutual information. So mutual information actually quantifies how much relative higher activity is tightly linked with a single behavior (or a very limited set of behaviors among many).

However, mutual information calculation can be biased especially in the context of limited data sets (Treves and Panzeri, 1995, *Neur Comp*; Wolpert and Wolf, 1995, *Phys Rev E*; Nemenman et al., 2004, *Phys Rev E*; Bonachela et al, 2008, *J Phys A*; Timme and Lapish, 2018, *eNeuro*). Indeed, we observed and reported in our initial submission (Fig. Supp. 7a-b, previously ED Fig. 6a-b) a bias of mutual information toward low active neurons. To circumvent this issue, we compared for each cell the actual value of mutual information to the distribution of mutual information calculated using shuffled distributions of the events (Ziv et al., 2013, *Nat Neurosci*; Danielson et al., 2017, *Neuron*; Stefanini et al., 2020, *Neuron*). As stated in the method section “detection of behaviour-active neurons”:

“For each cell, we generated 1000 random permutations of the events and calculated a behavior information value for each permutation, thus generating 1000 random behavior information values to which the actual behavior information value was compared. We labeled a cell as a behavior-active cell if the behavior information value was more than 4 sigma from the shuffled distribution”.

Concerning the detection of behavior-silent cells, we initially went for a similar shuffle approach to compare for each cell its actual activation occurrence value to the distribution of values obtained after a random permutation of calcium events. This approach yielded a detection of a very small proportion of “shuffle behavior-silent” dSPNs during most of the behaviors that precluded any further analysis of their behavior prediction capabilities (at least 20 neurons required). Because we wanted to be able to compare prediction capabilities between dSPNs and iSPNs for all the behaviors identified, we only reported results using a simple threshold approach (corresponding to the 95th percentile of activation occurrence values of “shuffle behavior-silent” neurons) in our initial manuscript. However, we agree

with Reviewer #3 and understand their concern. So in our revised manuscript, we incorporated the quantification of silent cells detected through a shuffle-based approach and reported behavior prediction results despite the absence of data for dSPNs for many behaviors (Supplementary Fig. 8b-e).

3. Several of the decoding results seem like they could be explained by how many neurons were used to decode rather than the identify of the neurons. For example, there are more “behavior active” D1 MSNs than there are D2 MSNs (4b), thus potentially explaining the results in 4c. Similarly, there are more “behavior silent” D2 MSNs than D1. Were matched numbers of cells used for decoding across all conditions?

As pointed by Reviewer #3, predictions using SVM can be affected by the number of cells used. We performed these controls before submitting our initial manuscript and, as we found that results were not qualitatively modified, we decided to not report them and use only data obtained without matched numbers of cells. However this is indeed a key control to actually reinforce our findings. We thus added a new supplementary figure (Supplementary Fig. 9) to illustrate that the decoding performances can increase with the number of cells used, and that our findings are not changed when similar number of cells are used between conditions for the predictions using behavior-active cells or behavior-silent cells.

4. It isn't clear why figures 3 and 4 use one measure of decoding, while figure 5 uses a different measure. This makes it difficult to compare the performance of behaviorally active vs behaviorally silent.

We agree with Reviewer #3 that the rational between our working hypotheses based on the selection-suppression model explained in the beginning of the results subsection “Neural code in iSPNs is biased toward silencing” and the use of a different decoding accuracy measure in this context was not explicit. We thus expanded this rational before reporting prediction-associated results as follow:

“We chose this measure because, according to the selection-suppression model¹⁷, consistent silencing in iSPNs should indicate that the animal experiences a given behavior, whereas their activation, leading to a suppression of this given behavior, should indicate that the animal experiences any other behavior without enabling to directly identify this other behavior.”

We would also like to point out that we did not attempt to compare the relative weights of behavior-active vs. behavior-silent; our goal with this study was to establish that the neural code of dSPNs is biased toward activation (a property not observed in iSPNs) and that the neural code of iSPNs is biased toward silencing (a property not observed in dSPNs).

5. The authors argue that D1 MSNs neural coding is biased towards activation while D2 MSNs are biased towards silencing. This may be true (with the all the caveats listed above), but we are talking about small differences. Forty percent of the D1 MSNs are not “behaviorally activated” and those neurons still perform above chance levels at decoding (4c). Similarly, for the most part, the small proportion of “behaviorally silent” D1 MSNs neurons don't perform at decoding any worse than their non-silent counterparts (and again seem to perform above chance levels). It was unclear how the authors accounted for these findings which seem counter to their model.

We understand reviewer's comment, as in our original manuscript we did not try to account for the relatively good decoding performances of “untuned” (either active or silent) neurons. A possible explanation could be that some neurons that are categorized as “untuned” are weakly tuned to

behaviors as they stand just below the threshold used for characterization. For example the distributions of behavior information or significance of behavior information are continuous and unimodal (Fig. 5b, Supplementary Fig. 7c). These weakly tuned cells still contain information most likely relevant for behavior prediction.

Moreover “untuned” neurons may also convey information through their shared ensemble activity patterns or their correlation with tuned neurons. The identification of such patterns would require investigating higher order models (secondary order or triplet correlation; e.g. Meshulam et al., 2017, Neuron). A significant contribution to the neural code of “untuned” neurons was found for example in the hippocampal formation (Stefanini et al., 2020, Neuron), in the prefrontal cortex (Leavitt et al., 2017, PNAS), or in the somatosensory cortex (Safaai et al., 2013, J Neurosci). This kind of analyses would of great use to try to better understand the neural code in the dorsal striatum alongside trying to identify whether “behavior-untuned” neurons could be significantly tuned toward any other external variable or if their activity is explained by a combination of external variables. We feel that these supplementary analyses are beyond the scope of the present study.

We included some additional text in the discussion to provide some explanation about the good decoding performances of “untuned” neurons.

6. The authors propose an interesting model (described above). If I have understood their model correctly, one would predict that for a given behavior A, average D2 MSN activity should be statistically lower than average for A, but statistically higher for all behaviors similar to A. The authors should test that prediction with the data they have.

We understand and agree with Reviewer’s hypothesis especially as we stated in the discussion that “highly similar behaviors are more likely to compete with each other than considerable different behaviors”. However, we would like to point that postulating a direct link between a “degree of competition” between behaviors and behavior distance (as evaluated by a difference in animals posture) is probably an oversimplification. For the reasons developed below, we updated this sentence in the discussion.

Indeed, according to our hypothesis (adaptive selection-suppression), behavior-active dSPNs should be strongly active during behavior A and behavior-silent iSPNs should be inactive during behavior A. Then during episodes of different behavior that is always competing with behavior A, behavior-active dSPNs and behavior-silent iSPNs (during behavior A) should both be strongly active as the expression of behavior A is suppressed (Figure 3a below). During episodes of a behavior that is never competing with behavior A, the same dSPNs and iSPNs should remain silent (no competition encoded in dSPNs activation, and no suppression required through iSPNs activation) (Figure 3a below). On average, for all episodes of a behavior that is often or rarely competing with behavior A, the “degree of competition” should be reflected in the amplitude of the average activity in both behavior-active during A dSPNs and behavior-silent during A iSPNs (i.e. the higher the competition, the higher the average activity) (Figure 3a below).

We thus computed the average activity of behavior-active dSPNs and behavior-silent iSPNs during the behavior for which they are tuned and compared it to other behaviors (during which they are not classified as behavior-active or behavior-silent, respectively) as a function of the behavior distance (Figure 3b-c below). For example, and following the formulation above, behavior-active dSPNs during locomotion straight (Figure 3b left panel) display a similar average activity during still sniffing, grooming, and rearing. This would indicate that still sniffing, grooming, and rearing compete with locomotion straight in a similar fashion despite largely different behavior distances between these behaviors and locomotion straight. Moreover, and supporting the above formulation, we observe that the average activity of behavior-silent iSPNs during locomotion straight is similar during still sniffing,

grooming, and rearing (Figure 3c left panel). The opposite seems true with pairs of behaviors displaying a similar behavior distance and different average activities as exemplified by the comparison of the average activities of behavior-active dSPNs or behavior-silent iSPNs during locomotion turn left vs. locomotion fast, still turn right, or still sniffing (Figure 3b and 3c middle panels). Altogether, these observations indicate that the behavior distance is not a good measure of how often two behaviors may be competing with each other making it complicated to test any prediction about the level of activity of behavior-active dSPNs and behavior-silent iSPNs during other behaviors.

Figure 3. Activity of behavior-active dSPNs and behavior-silent iSPNs during other behaviors.

a, Diagram representing the predicted co-evolution of neuronal activity of behavior-active dSPNs and behavior-silent iSPNs during different behaviors as a function of how often they compete with each other. **b-c**, Examples of the average activity of behavior-active dSPNs (**b**) and behavior-silent iSPNs (**c**) during the behavior for which they are tuned and compared to other behaviors as a function of the behavior distance. **d**, Coevolution of the average activity of behavior-silent iSPNs and behavior-active dSPNs during all the behaviors (color coded). The left rectangle groups together observations for which dSPNs are categorized as behavior-active and iSPNs are categorized as behavior silent. The right rectangle corresponds to observations during other behaviors.

We also observed, for example for behavior-silent iSPNs during grooming (Figure 3c right panel), that the average activity of these behavior-silent iSPNs increases with behavior distance in particular for locomotor behaviors. This observation could be linked with the difference in average activity of behavior-active dSPNs during behaviors for which they are tuned. Indeed, when we plotted the activity of behavior-silent iSPNs against the activity of behavior-active dSPNs for all behaviors (Figure 3d), we observed that the activity of behavior-active dSPNs is not similar for all behaviors

(Figure 3d left rectangle), and that the activity of behavior-silent iSPNs during other behaviors is larger during behaviors characterized by a higher activity of behavior-active dSPNs. It seems like the activation of behavior-silent iSPNs during other behaviors scales with the activity of behavior-active dSPNs during the ongoing behavior.

We are still trying to clearly understand these observations. However, for now, we are lacking a proper dedicated measure of competition between pairs of behaviors that may also change as a function of the external and/or internal context. A possible solution to better investigate this point would be to record animals in a more tightly controlled environment in which the set of accessible/competing behaviors is strongly restricted. Also, because understanding this point would likely require analyzing neuronal activity on an episode per episode basis (to acutely probe the set of activations in dSPNs encoding competing behaviors, and activations in iSPNs encoding suppression of these competing behaviors), we think that recording both dSPNs and iSPNs simultaneously with a dual-color imaging system would be critical to test further predictions.

7. In extended figure 4, please show the effect of amphetamine on firing rates for D1 and D2 MSNs.

According to reviewer's request, we added in Supplementary Fig. 5 the effects of saline and amphetamine injections on the average population activity recorded in dSPNs and iSPNs (Supplementary Fig. 4c,d).

Minor

1. Why don't the values on lines 184-185 match those in figure 3b?

The values reported in Figure 4b (formerly Figure 3b) correspond to the accuracy to predict the correct behavior among any of the 12 behaviors using the combined output of all the 66 individual binary classifiers. The values reported on lines 184-184 correspond to the average accuracy of each binary classifier, i.e. the accuracy to predict the correct behavior between two behaviors. These values correspond to the y-axis in Figure 3d. We made some additions in the corresponding result section to clarify this distinction.

2. Overall, the manuscript is very well-written. There are some typos in the figure legend (line 630, 633 and 793 should read 'blue')

Thank you for pointing these mistakes.

3. Does the color coding for the behavior have numerical meaning in terms of distance? If so, it should be labeled accordingly.

The color palette for behaviors does not correspond to any distance measure. Colors were chosen by spiraling through hue and saturation and increasing brightness.

4. Some of the statistical legends were unclear. For example, in extended figure 4d, the D2 MSN and shuffle reconstruction bars post amphetamine have nearly identical values but are listed as having ### p<0.001. Is this correct? But there are also lines with astrixes that have associated different values...

In these panels (now labeled Supplementary Fig 5e-g), there are three different post-hoc comparisons illustrated:

++ signs correspond to comparisons between recorded data and shuffles

signs correspond to comparisons between saline and amphetamine

** signs correspond to comparisons between dSPNs and iSPNs

The significant difference mentioned by the reviewer corresponds to a comparison between the prediction accuracy after saline injection and the prediction accuracy after amphetamine injection. However, we agree that it can be difficult to identify each symbol and its meaning. So we added at the bottom of the figure some text to provide symbols meaning.

REVIEWERS' COMMENTS

Reviewer #1 (Remarks to the Author):

This manuscript, which was already at a high level upon initial submission, has further improved following incorporation of the reviewers' comments. I find the manuscript acceptable in its current form.

I do, however, suggest the authors address the following issues:

Line 90-91 – ‘as a result, the coactivation of specific subsets of dSPNs and iSPNs would result in the selection and execution of only one motor program’ – this statement follows the statement that patterns of dSPN activity are associated with (‘encode’) specific behaviors, while the converse inhibition is observed for iSPNs.

Did the authors want to relate to specific dSPN activation, coincident with specific iSPN inhibition?

Line 92-94 - “This updated model...”. In what way is the model updated? What is the ‘new’ model the authors want to propose? The authors provide a clear description of their model in Fig. S12 (‘adaptive selection-suppression’), I suggest that they allude to the model and provide a brief explanation of its workings at this point in the text.

Behavioral annotation:

The added information about the methodology used to annotate behavior is crucial to the evaluation of the manuscript. Overall the approach is logical and appears to be reasonably efficacious. However, the fact that immobility is scored with low efficacy is a matter of slight concern. This should be the easiest category to score – as locomotion is stopped. It is also worthy of discussion that the majority of the behavioral categories were scored at <60% efficiency by the automated scoring system, while only 3/11 categories (locomotion straight & locomotion right & locomotion left) appeared to score at >80% efficacy. I would expect these 3 categories to be scored at close to 100% efficacy from a simplified analysis of locomotion trajectories with no need for a complex model.

The low efficacy of the system is expected to cap the capacity of the authors to make effective associations of behavioral annotation to neural activity, unless ‘similar’ behavioral categories (i.e. those categories more prone to be confused for each other by the behavioral annotation algorithm) are relatively similar in their activity signatures. The data presented by the authors, especially in figure S6, is consistent with this notion, as similar behaviors appear to cluster in the similarity of their neural activity signatures. To me, this raises the possibility that activity in the dorsal striatum encodes the propensity to engage in broad action categories, rather than in a specified action with well defined dynamics. Perhaps

the precise specification of actions from broad categories occurs downstream of the striatum? The authors may want to discuss this general question.

Reviewer #2 (Remarks to the Author):

This remains an ambitious, large, and complex study. The authors have filled in many missing details about the methodology and improved the presentation. I'm not sure that I agree with all of the choices they made in the experimental pipeline. The need for manual adjustments to the analysis and for multiple runs followed by consensus voting making me a bit uneasy. However, the results seem to be on solid ground and the only way to have these pipelines and conclusions vetted by the community is to publish the paper and the methods.

Reviewer #3 (Remarks to the Author):

After reading the authors response, I still think that a permutation test could have been used to determine behaviorally activated neurons (just as the authors were able to do for behaviorally silenced neurons) and that this result would add to the manuscript (regardless of outcome). After all, the MI measure seems biased toward the more predictable dSPNs.

Apart from this point, I believe the authors have addressed my concerns and congratulate them on their substantive manuscript.

REVIEWERS' COMMENTS

Reviewer #1 (Remarks to the Author):

This manuscript, which was already at a high level upon initial submission, has further improved following incorporation of the reviewers' comments. I find the manuscript acceptable in its current form.

We would like first to thank the Reviewer for considering the valuable corrections we made to the initial version of our manuscript thanks to their first series of comments.

I do, however, suggest the authors address the following issues:

Line 90-91 – ‘as a result, the coactivation of specific subsets of dSPNs and iSPNs would result in the selection and execution of only one motor program’ – this statement follows the statement that patterns of dSPN activity are associated with (‘encode’) specific behaviors, while the converse inhibition is observed for iSPNs.

Did the authors want to relate to specific dSPN activation, coincident with specific iSPN inhibition?

Line 92-94 - “This updated model...”. In what way is the model updated? What is the ‘new’ model the authors want to propose? The authors provide a clear description of their model in Fig. S12 (‘adaptive selection-suppression’), I suggest that they allude to the model and provide a brief explanation of its workings at this point in the text.

We understand Reviewer’s comments above highlighting that the concluding sentences of our introduction were inadequately organized and quite difficult to follow. As a consequence we rewrote and reorganized this part to explain more clearly our conclusions and the model we infer from our observations.

Behavioral annotation:

The added information about the methodology used to annotate behavior is crucial to the evaluation of the manuscript. Overall the approach is logical and appears to be reasonably efficacious. However, the fact that immobility is scored with low efficacy is a matter of slight concern. This should be the easiest category to score – as locomotion is stopped. It is also worthy of discussion that the majority of the behavioral categories were scored at <60% efficiency by the automated scoring system, while only 3/11 categories (locomotion straight & locomotion right & locomotion left) appeared to score at >80% efficacy. I would expect these 3 categories to be scored at close to 100% efficacy from a simplified analysis of locomotion trajectories with no need for a complex model.

We understand Reviewer’s regard to our pipeline efficacy/efficiency. A rapid survey of some of the most recent published behavior identification pipeline reveals that our pipeline performs with a relatively similar efficacy. For example, DeepEthnogram (Bohnslav et al., 2021, eLife) detects locomotion with a F1-score of about 0.9, whereas our pipeline detect locomotion straight with a F1-score of 0.91 (0.81 for locomotion with right turn and 0.88 for locomotion with left turn; Supplementary Fig. 2i). DeepAction (Harris et al., 2023, Scientific Reports) detects moving and rear

with F1-scores of about 0.7 and 0.65 respectively. Our pipeline has a F1-score for rearing of 0.67 (Supplementary Fig. 2i). The overall F1-score for DeepAction is about 0.7, similar to our pipeline (Supplementary Fig. 2k). B-SOiD (Hsu and Yttri) detects locomotion, rear, and inactive classes with around 99%, 92% and 88% accuracy, respectively (F1-scores not reported). With our pipeline, we detect locomotion straight, rearing, and immobility with 98%, 93% and 91% accuracy, respectively. Importantly, we showed that the detection performance of our pipeline for different behaviors correlates with the intra-annotator consistency in labeling behaviors (Supplementary Fig. 2f). It indicates that some behaviors that are harder to classify for human annotators are also harder to separate with the automatic pipeline. We think that currently the main limiting factor in improving detection capabilities lies in the quality of the videorecording itself. Most of the confusion for both human annotators and the algorithm occur for quiet behaviors (e.g. rooming, still sniffing, immobility) that most likely comes from recordings performed from the top of the animal. We think that, using an additional video stream recorded for example with a camera placed below the animal under a transparent floor could greatly improve the distinction between these behaviors.

The low efficacy of the system is expected to cap the capacity of the authors to make effective associations of behavioral annotation to neural activity, unless 'similar' behavioral categories (i.e. those categories more prone to be confused for each other by the behavioral annotation algorithm) are relatively similar in their activity signatures. The data presented by the authors, especially in figure S6, is consistent with this notion, as similar behaviors appear to cluster in the similarity of their neural activity signatures. To me, this raises the possibility that activity in the dorsal striatum encodes the propensity to engage in broad action categories, rather than in a specified action with well defined dynamics. Perhaps the precise specification of actions from broad categories occurs downstream of the striatum? The authors may want to discuss this general question.

We agree with Reviewer #1 that the theoretical framework they describe above could explain some experimental observations we reported in this manuscript as well as other experimental evidence reported before. Thus we added some sentences in the discussion to mention this possible interpretation. However, we feel that a more detailed, and probably necessary, comparison between the various theoretical views and experimental data would more appropriate in a review-style article.

Reviewer #2 (Remarks to the Author):

This remains an ambitious, large, and complex study. The authors have filled in many missing details about the methodology and improved the presentation. I'm not sure that I agree with all of the choices they made in the experimental pipeline. The need for manual adjustments to the analysis and for multiple runs followed by consensus voting making me a bit uneasy. However, the results seem to be on solid ground and the only way to have these pipelines and conclusions vetted by the community is to publish the paper and the methods.

We would like to express our gratitude to the Reviewer for their positive feedback and for the valuable revisions they enable us to incorporate into our manuscript.

Reviewer #3 (Remarks to the Author):

After reading the authors response, I still think that a permutation test could have been used to determine behaviorally activated neurons (just as the authors were able to do for behaviorally silenced neurons) and that this result would add to the manuscript (regardless of outcome). After all, the MI measure seems biased toward the more predictable dSPNs.

Apart from this point, I believe the authors have addressed my concerns and congratulate them on their substantive manuscript.

First we would like to extend our appreciation to the Reviewer for acknowledging the significant improvements we made to the initial version of our manuscript based on their initial feedback.

We are not sure we fully understand Reviewer's reservations concerning the mutual information measure, as it is indeed a metric designed to quantify how much information is shared between two signals. Anyway, we agree that what Reviewer #3 suggests can constitute an additional valuable control. We thus added a quantification of neurons significantly active during behaviors using a shuffle procedure and an evaluation of decoding performance using these neurons in Supplementary Fig. 12.